# Hyperspherical Simplex Representations from Softmax Outputs and Logits are Inherently Backward-Compatible

## Abstract

Training modern AI models has become increasingly expensive, and updating deployed models can significantly alter the behavior of applications built upon them, due to changes in internal feature representations. In retrieval systems, such updates often require re-indexing the gallery-set by extracting feature vectors for all gallery data, a process that is computationally expensive and time-consuming, especially for large-scale datasets. Existing backward-compatibility methods allow direct comparison between the representations of updated and old models, avoiding the re-indexing of the gallery. However, they typically introduce a dependency on the old model by using auxiliary losses, mapping functions, or specific modifications to the model architecture. In this paper, we show that independently trained models are inherently backward-compatible when hyperspherical simplex representations derived from their softmax outputs or logits are used. Leveraging the geometric structure induced by the softmax function on these features, we define a deterministic linear transformation that preserves their alignment across model updates. We demonstrate that these representations satisfy in expectation the formal definition of backward-compatibility. Without relying on regularization-based training, mapping functions, or modifications to the model architecture, we achieve competitive results on standard compatibility benchmarks involving model updates with new training classes and/or advanced model architectures.

## 1 Introduction

Training modern AI models has become increasingly expensive, limiting accessibility to a few well-resourced organizations (Wolf et al., 2019; Radford et al., 2021; Dubey et al., 2024; Jiang et al., 2023; Anthropic, 2024). Despite reductions in model parameters and computing costs (Dubey et al., 2024), training from scratch, fine-tuning, and inference remain economically challenging (Wang et al., 2025; OpenAI, 2024). As a result, many models are being released in public repositories (Marcel & Rodriguez, 2010; Wolf et al., 2019; Wightman, 2019) or offered as services through APIs, a trend likely to continue given the persistent high training costs and the performance benefits outlined by scaling laws (Kaplan et al., 2020). This shift towards releasing models in public repository and providing them via APIs has not only simplified the development of new applications but also enabled their widespread use across various fields.

As these models become more integrated into real-world systems, the need to update them is also growing. This demand is driven by several factors (Raffel, 2023; Yadav et al., 2025), including evolving training strategies (Biondi et al., 2024; Echterhoff et al., 2024; Shen et al., 2020), advancements in model architectures (Touvron et al., 2023), the availability of higher-quality datasets (Gunasekar et al., 2023), the expansion of training classes, or extended training periods (Biderman et al., 2023; Raffel, 2023). Such updates not only improve performance but also consolidate the rapid progress of the field into more usable, unified models—simplifying deployment and adoption despite the continued growth in scale and complexity of models, datasets, and infrastructures (Bommasani et al., 2021; Sorscher et al., 2022). However, a significant drawback of model updates is that they can compromise compatibility with earlier versions, altering the behavior of applications or services built upon them (Raffel, 2023). For example, image retrieval services might experience unexpected changes in image rankings following a model update, forcing users to adapt when previously top-ranked images no longer appear first (Shen et al., 2020). Similarly, end users, such as

drivers of semi-autonomous cars, often develop a mental model of the machine learning system's capabilities (Bansal et al., 2019a). When updates modify system behavior, users must adjust their mental model of the system's functionality, a task that can be challenging, dissatisfying, and even unsafe (Bansal et al., 2019b; Templeton, 2019). Recent observations have highlighted similar issues with large language models (LLMs) (Echterhoff et al., 2024; Goswami et al., 2025), raising concerns about AI safety and alignment. This is particularly critical when decisions and textual outputs from these agents are translated into actions (Amodei et al., 2016; Ngo et al., 2024). In this case, it is essential that the predictions of the updated model mimic those of the old model for any correct classification (Milani Fard et al., 2016; Yan et al., 2021).

Updating a model while ensuring compatibility with its earlier versions has been explored along three complementary dimensions: 1) Weights-level, which explores the merging of model parameters under the assumption of linear mode connectivity, to unify multiple versions into a single, robust model (Ainsworth et al., 2023; Wortsman et al., 2022; Frankle et al., 2020; Marczak et al., 2026); 2) Representation-level, which seeks to learn backward-compatible representations that are comparable across model updates, so they can be used interchangeably in time (Shen et al., 2020; Biondi et al., 2023a); and 3) Classifier decision-level, which ensures that the updated model preserves the correct predictions of the old model, preventing negative flips (Milani Fard et al., 2016; Yan et al., 2021; Zhao et al., 2024; Echterhoff et al., 2024; Ricci et al., 2026).

Among these paradigms, representation-level compatibility demonstrates particular relevance in retrieval systems. Backward-compatibility training, firstly introduced by Shen et al. (2020) to avoid image re-indexing of the gallery-set, has been established as a standard training paradigm to achieve backward-compatibility between model representations. However, most existing studies on backward-compatibility require training a new model using the old model as a reference (Shen et al., 2020; Meng et al., 2021; Duggal et al., 2021; Pan et al., 2023; Zhou et al., 2023). This approach has become increasingly challenging due to restricted access to old model architectures and weights—often limited by proprietary constraints and non-disclosure policies—as well as the high computational costs associated with training large-scale modern AI models. Recent studies (Biondi et al., 2023a; 2024) have addressed this issue leveraging models that are compatible even if they have been trained without explicitly using the previous model as a reference. However, their implementation requires agreement on using a specific pre-allocated fixed classifier among parties before training the models (Pernici et al., 2019a; 2021a; Zhu et al., 2021). This agreement is often challenging due to competition-related issues between organizations, and the fixed classifier requires substantial pre-allocation for future classes.

In this paper, we show that independently trained models, i.e., without explicitly using the previous model as a reference, are inherently backward-compatible by using hyperspherical simplex representations derived from their softmax outputs or logits. Our formulation leverages the simplex geometry induced during training by the softmax function and one-hot encoded class labels as a fixed, model-agnostic reference, yielding hyperspherical simplex representations that are inherently aligned across model updates. Although their simplex geometries evolve with the introduction of new classes, we propose a deterministic linear transformation to compensate for the angular misalignment between the old and new simplex geometries, thereby preserving representation alignment across updates. We analytically demonstrate that these representations satisfy, in expectation, the formal definition of backward-compatibility as defined by Shen et al. (2020) due to their geometric properties.

However, this approach introduces a scalability challenge: since the representations are derived from model outputs, their dimensionality scales linearly with the number of training classes, resulting in a substantial memory footprint. To address this, we exploit the top-$k$ calibration property of the softmax function (Lapin et al., 2016; Yang & Koyejo, 2020), which preserves the ranking of the probabilities. It enables top-$k$ sparsification, significantly reducing the storage overhead of the gallery features with negligible retrieval performance degradation.

Our main contributions are the following:

1. We introduce a novel approach to achieve backward-compatibility that leverages hyperspherical simplex representations derived from softmax outputs and logits.

2. We provide theoretical analysis demonstrating that these hyperspherical simplex representations satisfy, in expectation, the formal definition of backward-compatibility due to their inherent geometric configuration.

3. We propose a top-$k$ sparsification method for hyperspherical simplex representations leveraging the top-$k$ calibration property of the softmax function, which enables a theoretically grounded feature dimensionality reduction.

4. We achieve competitive results on standard compatibility benchmarks, obtaining significant improvements in scenarios involving a high number of model updates, without using additional losses, mapping functions, or modifications to the model architecture. We provide empirical evidence that features derived from softmax outputs and logits provide this compatibility, even between independently trained models available in a public repository.

The remaining part of the paper is organized as follows. In Sec. 2, we review related work on backward-compatibility. In Sec. 3, we derive hyperspherical simplex representations from softmax outputs and logits, and demonstrate that they satisfy the formal definition of backward-compatibility. In Sec. 4, we present extensive experiments and provide comparative analysis between our proposed method and state-of-the-art alternatives. Finally, in Sec. 5, we draw conclusions and discuss the limitations of this work.

## 2 Related Works

Backward-compatibility was first formalized by Shen et al. (2020), who defined both theoretical and empirical criteria to verify whether learned representations are interoperable across model updates, thus eliminating the need to re-index the gallery-set in retrieval systems. To learn backward-compatible representations, they proposed Backward-Compatible Training (BCT), where the previously learned classifier is frozen and serves as a fixed reference during training of the new model, aligning new feature vectors with their corresponding old class prototypes. Subsequent work proposed additional regularization objectives to enforce alignment between old and new representations (Budnik & Avrithis, 2021; Duggal et al., 2021; Meng et al., 2021; Zhang et al., 2021; 2022; Pan et al., 2023; Wu et al., 2022; Goswami et al., 2025). In Meng et al. (2021), a combination of loss functions is used to align class means between different model upgrades and to achieve more compact intra-class distributions while learning new information, directly optimizing a point-to-set formulation of backward-compatibility. In Zhang et al. (2021), compatible representations are learned with a contrastive loss that encourages each new-to-old positive pair to be more similar than both new-to-old and new-to-new negative pairs. An adversarial learning discriminator is introduced in BCT by Pan et al. (2023) to minimize the distribution gap between features from the old and new models. In these regularization-based approaches, due to the constraints imposed to achieve compatibility, the performance of the new backward-compatible model often does not reach that of a newly independently trained model (Zhou et al., 2023), trained without using the old model as reference. To address this issue, Zhou et al. (2023) and Ricci et al. (2024) focus on modifying the model architecture by expanding the feature space at each model update, such that the original subspace preserves compatibility with previous models, while information from the new data is learned in the additional dimensions. However, even if compatibility is achieved while retaining the discriminative performance of a newly independently trained model, these approaches require explicit modifications to the model architecture that introduce growing complexity with each successive feature space expansion.

On a different line, several studies have proposed mapping-based approaches to achieve compatibility across model updates (Chen et al., 2019; Wang et al., 2020; Meng et al., 2021; Ramanujan et al., 2022; Hu et al., 2022; Jaeckle et al., 2023; Ricci et al., 2025). These methods learn mapping functions to align corresponding representations of the same inputs obtained from two different models, enabling direct comparison between them. These mapping-based methods are closely related to prior work in cross-model and cross-modal representation learning, where alignment between feature spaces of different modality is a central objective (Moschella et al., 2023; Cannistraci et al., 2023; 2024; Maniparambil et al., 2024; 2025). However, while such approaches can, in principle, achieve compatibility between independently trained models, they introduce significant overhead in scenarios with many model updates, as they necessitate composing multiple

mapping functions across successive updates. To address these limitations, the solution proposed by Biondi et al. (2023a;b; 2024) involves learning with a pre-allocated $d$-simplex fixed classifier, which enforces feature alignment across updates by fixing the position of class prototypes a priori (i.e., since the beginning of training), while accommodating new classes within pre-allocated regions of the feature space. Although this approach achieves compatibility without using the old model as reference, it relies on the same pre-allocated fixed classifier across model updates and requires architectural modifications.

Unlike prior work, we demonstrate that independently trained models are inherently compatible by leveraging the properties of the softmax function to derive hyperspherical simplex representations from the softmax outputs and logits. These representations are aligned across model updates, thus eliminating the need for additional losses, mapping functions, or modifications to the model architecture to achieve compatible representations.

## 3 Compatibility of Hyperspherical Simplex Representations

In this section, we demonstrate that the hyperspherical representation derived from softmax outputs and logits satisfies the backward-compatibility definition in Shen et al. (2020).

### 3.1 Preliminaries and Motivations

According to Shen et al. (2020), backward-compatibility across model updates is achieved when the representation of an updated model learned at step $t$ can be compared with the representation of the earlier model learned at step $k$. Formally, it requires that the following two inequalities hold:

**Definition 3.1** (**Backward-Compatibility**)**.**

$$d\big(\mathbf{h}_i^t, \mathbf{h}_j^k\big) \leq d\big(\mathbf{h}_i^k, \mathbf{h}_j^k\big), \forall (i,j) \in \{(i,j) \mid y_i = y_j\} \tag{1a}$$

$$d\big(\mathbf{h}_i^t, \mathbf{h}_j^k\big) \geq d\big(\mathbf{h}_i^k, \mathbf{h}_j^k\big), \forall (i,j) \in \{(i,j) \mid y_i \neq y_j\} \tag{1b}$$

where $\mathbf{h}^t$ and $\mathbf{h}^k$ are feature vectors extracted respectively with the updated model at step $t$ and the earlier model at step $k$, $d$ a distance function (e.g., cosine distance), and $y_i$ and $y_j$ are class labels associated to $\mathbf{h}_i$ and $\mathbf{h}_j$, respectively.

The inequalities in Eq. 1a and Eq. 1b specify that the updated model representation, when used to compare against the old representation, should perform at least as well as the old one in grouping images from the same classes and separating those from different classes.

While Eq. 1a and Eq. 1b describe desirable goals for compatible representations, they require an exhaustive comparison of the two representations, which is challenging in practice (Shen et al., 2020). In recent work, Biondi et al. (2024) study backward-compatibility using representations as learned according to a $d$-simplex fixed classifier. In their analysis, class features are modeled as uniformly distributed Euclidean hyperballs. Under this assumption, Euclidean distances between features are unbounded, and inter-class separation depends jointly on the displacement of class centers and the radii of the hyperballs. They show empirically that Eq. 1a is satisfied when clusters of features from the same class remain aligned across model updates, whereas Eq. 1b is not. Their conclusion is therefore specific to this Euclidean hyperball formulation.

In contrast, we study hyperspherical simplex representations under a von Mises-Fisher (vMF) assumption on the unit hypersphere, where squared Euclidean distance is proportional to cosine distance and is therefore bounded. Under this geometry, inter-class separation is governed by angular relations between class prototypes. In this setting, we analytically demonstrate that hyperspherical simplex representations derived from softmax outputs and logits satisfy in expectation both conditions of Def. 3.1, stemming from two key properties of these representations: (a) alignment across successive model updates; and (b) an inter-class angular separation greater than $\pi/2$. As formally shown in Theorem 3.6, property (b) is precisely the condition required to satisfy Eq. 1b. Since these properties are an inherent byproduct of training with softmax cross-entropy loss, our approach provides a practical and theoretically grounded path to achieve backward-compatibility between independently trained models.

Importantly, our theoretical result is not exclusive to our approach: it extends to any regular simplex representation whose geometry remains aligned across updates, including the one learned through fixed $d$-simplex classifiers such as those considered by Biondi et al. (2023a; 2024) (see Remark 1). The main distinction from Biondi et al. (2023a; 2024) is that hyperspherical simplex representations derived from softmax outputs and logits do not require a fixed classifier to be specified a priori.

## 3.2 Hyperspherical Simplex Representations

We now formalize two hyperspherical simplex representations, which are derived directly from the softmax outputs and logits of a trained classification model.

### 3.2.1 Features Derived From Softmax Outputs

Given a model at time step $k$, the softmax output $\sigma(\mathbf{z}^k)$, obtained by applying the softmax function $\sigma(\cdot)$ to the logits $\mathbf{z}^k$, represents a probability distribution over the $C^k$ classes and lies within the probability simplex, which is defined as follows:

**Definition 3.2** (**Probability Simplex**). The probability simplex $\Delta^{C^k-1}$ is the $(C^k-1)$-dimensional simplex in $\mathbb{R}^{C^k}$ whose vertices are the canonical basis vectors $\mathbf{e}_1, \mathbf{e}_2, \ldots, \mathbf{e}_{C^k}$, and its center is $\mathbf{o}^k = \frac{1}{C^k} \sum_{c=1}^{C^k} \mathbf{e}_c$, such that $\Delta^{C^k-1} = \{\mathbf{u} \in \mathbb{R}^{C^k} : u_1 + \cdots + u_{C^k} = 1, \ u_j \geq 0 \ \ for \ \ j = 1, \ldots, C^k\}$.

Assuming the model has sufficient capacity to map training data to a linearly separable representation space (Soudry et al., 2018; Fang et al., 2021), the optimization of the cross-entropy loss leads softmax outputs to converge toward the vertices $e_c$ of the probability simplex $\Delta^{C^k-1}$ during training, as the target one-hot encoded labels correspond to these vertices (see Fig. 1). To derive hyperspherical representations from softmax outputs, the $\ell_2$-normalization is the most intuitive approach. However, this operation entails a critical geometric limitation for compatibility. Since the probability simplex $\Delta^{C^k-1}$ resides entirely within the positive orthant, its center $\mathbf{o}^k$ is offset from the coordinate origin. Consequently, $\ell_2$-normalized softmax outputs exhibit inter-class angles strictly smaller than $\pi/2$, thereby failing to satisfy Eq. 1b as established in the analytical proof of Theorem 3.6 (see Appendix A).

To obtain pairwise angles between distinct classes exceeding $\pi/2$, we introduce a projection matrix $\mathbf{P}_{k,k}$ before the $\ell_2$-normalization of the softmax outputs. The projection matrix translates the center of the probability simplex $\Delta^{C^k-1}$ to the origin, projecting the softmax outputs onto a $(C^k - 1)$-regular

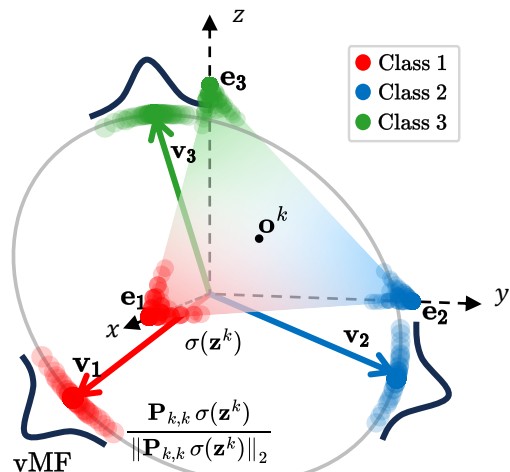

Figure 1: Softmax outputs form a probability simplex (colored area) centered at $\mathbf{o}^k$. The projection $\mathbf{P}_{k,k}$ shifts the probability simplex center to the origin, projecting softmax outputs into a regular simplex. The features $\mathbf{h}^k$ of Eq. 4 are then obtained through normalization onto the hypersphere. For each class, features closely approximate a von Mises-Fisher distribution centered on the class prototypes (colored vectors).

simplex[1] which retains the geometry of the original probability simplex, as shown in Fig. 1.

To formulate the projection matrix, we define the class prototypes $\mathbf{v}_c^k$ as the vectors representing the directions from $\mathbf{o}^k$ to the vertices $\mathbf{e}_c$ to which the softmax outputs align:

$$\mathbf{v}_c^k = \mathbf{e}_c - \mathbf{o}^k \quad \forall c, \ 1 \leq c \leq C^k, \quad \mathbf{v}_c^k \in \mathbb{R}^{C^k}. \tag{2}$$

---

[1]A $(C^k - 1)$-regular simplex is a simplex centered at the origin where all vertices are equidistant and the angle between any pair of vertices is constant, specifically $\arccos(-1/C^k - 1)$.

Since the class prototypes $\mathbf{v}_c^k$ are derived from fixed canonical basis vectors $\mathbf{e}_c$, they maintain the same positions in the feature space across model update, thus providing an inherent alignment of hyperspherical simplex representations. Furthermore, they form a regular simplex centered at the origin, exhibiting an angular separation greater than $\pi/2$ (Pernici et al., 2021a). The set of these vectors spans the $(C^k - 1)$-dimensional hyperplane of the regular simplex, a fundamental property that allows these vectors to directly define the subspace for the projection. Thus, the deterministic projection $\mathbf{P}_{k,k}$ is defined as the following $C^k \times C^k$ matrix:

$$\mathbf{P}_{k,k} = \mathbf{V}^k = [\mathbf{v}_c^k]_{c=1}^{C^k} = \mathbf{I}_{C^k} - \frac{1}{C^k}\mathbf{J}_{C^k} \tag{3}$$

where $[\mathbf{v}_c^k]_{c=1}^{C^k}$ denotes the matrix obtained by stacking the class prototype vectors $\mathbf{v}_c^k$ as columns, which is equivalent to the centering matrix as described by Marden (1996), with $\mathbf{I}_{C^k}$ the identity matrix for $C^k$ classes and $\mathbf{J}_{C^k}$ a $C^k \times C^k$ matrix entirely composed of ones. The hyperspherical simplex representation $\mathbf{h}^k$ is the $\ell_2$-normalized projection[2] of softmax output vector via $\mathbf{P}_{k,k}$, defined as:

$$\mathbf{h}^k = \frac{\mathbf{P}_{k,k}\,\sigma(\mathbf{z}^k)}{\|\mathbf{P}_{k,k}\,\sigma(\mathbf{z}^k)\|_2}, \quad \mathbf{h}^k \in \mathbb{R}^{C^k}. \tag{4}$$

On the hypersphere, each class cluster of the hyperspherical simplex representation can be statistically modeled by a von Mises–Fisher (vMF) distribution (Fisher, 1953; Scott et al., 2021), the hyperspherical analogue of the Gaussian distribution (see Fig. 1), where the mean direction of each vMF coincides with the class prototype $\mathbf{v}_c$. This allow us to analytically characterize both Eq. 1a and Eq. 1b through the closed-form solution employed in Theorem 3.6 proof, thereby demonstrating that feature representations $\mathbf{h}^k$ of Eq. 4 satisfy the compatibility definition.

### 3.2.2 Features Derived From Logits

Minimizing the cross-entropy loss on softmax outputs constrains the underlying logits via the softmax function $\sigma(\cdot)$, inducing a geometric correspondence between the softmax outputs distribution and the logit distribution. Consequently, the logits form a simplex configuration characterized by the class prototypes defined in Eq. 2. We formalize this relationship in the following proposition:

**Proposition 3.3** (Proof in Appendix B). *During training, while softmax outputs converge to the vertices of the probability simplex, the corresponding logits vectors align to directions as defined in Eq. 2.*

Fig. 2 provides intuition of Prop. 3.3 in a representative case with three classes of CIFAR-10 dataset. At the end of training, the softmax outputs concentrate near the vertices of the probability simplex (colored area), while the logits arrange in a regular simplex (gray area) with same class prototypes $\mathbf{v}_c^k$ of the representation derived from the softmax outputs in Eq. 4. This observation motivates the

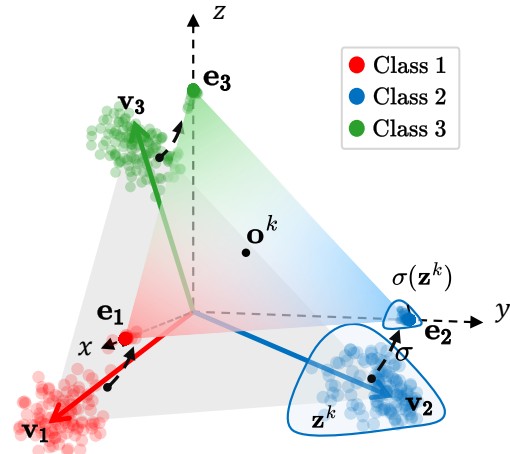

Figure 2: Softmax outputs and logits obtained from a ResNet-18 trained on three CIFAR-10 classes. While softmax outputs concentrate near the vertices of the probability simplex (colored area), logits form a similar simplex geometry (gray area) centered at the origin with a broader distribution spread.

formulation of a second hyperspherical simplex representation, which exhibits similar alignment properties for class prototypes and maintains the same class separation as the previous one of Eq. 4. Unlike the probability simplex, the logit simplex is centered at the origin, as illustrated in Fig. 2. Consequently, logits can

---

[2]When the softmax output corresponds to the uniform distribution (i.e., maximum predictive uncertainty) so that $\sigma(\mathbf{z}^k) = \mathbf{o}^k$, the projected vector $\mathbf{P}_{k,k}\,\sigma(\mathbf{z}^k)$ is the zero vector, yielding the $\ell_2$-normalization undefined. In this particular case, we define $\mathbf{h}^k \triangleq \mathbf{0}$, reflecting null similarity with all class prototypes. It is worth noticing that, while this case is theoretically possible, as trained classifiers typically produce non-uniform output distributions (Hendrycks & Gimpel, 2017).

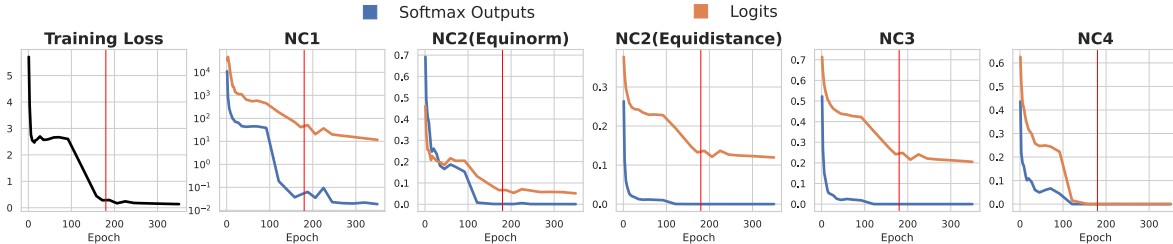

Figure 3: Plots of training loss and Neural Collapse hypotheses for softmax outputs (blue) and logits (orange) on CIFAR-100 with ResNet-18. The terminal phase of training is shown with a vertical red line.

be directly $\ell_2$-normalized, resulting in the following hyperspherical simplex representation:

$$\mathbf{h}^k = \frac{\mathbf{z}^k}{\|\mathbf{z}^k\|_2} \ , \quad \mathbf{h}^k \in \mathbb{R}^{C^k}. \tag{5}$$

Analogously to Eq.4, this representation can also be statistically modeled with a vMF distribution (Scott et al., 2021) and thus analyzed analytically in Theorem 3.6. However, the logits-based representation exhibits a larger angular spread than features of Eq. 4 (see Fig. 2), which improves feature generalization and robustness in open-set scenarios (Wang & Isola, 2020). To empirically verify Prop. 3.3 and the spread of features, we leverage the Neural Collapse (NC) hypotheses introduced by Papyan et al. (2020). These hypotheses provide a methodology for examining how softmax outputs and logits collapse to the class prototypes defined in Eq. 2. The Neural Collapse hypotheses evaluate: (a) whether the within-class covariance tends to 0 (NC1); (b) whether class means of features tend to form a simplex whose vertices have same norm (NC2 equinorm) and are placed at the maximum possible distance from each other (NC2 equidistance); (c) whether class means converge to class prototypes (NC3); whether the nearest class-means rule for classifying features holds (NC4). Fig. 3 shows the NC hypotheses values for both softmax outputs (blue curves) and logits (orange curves) at the terminal phase of training for a ResNet-18 on CIFAR-100. From the NC1 plot, we observe that the orange curve is higher than the blue curve, indicating that logits present a greater within-class spread than softmax outputs. Moreover, in the NC2 equidistance and NC3 plots, the orange curves are also higher than the blue curves. This shows that logits exhibit a lower degree of alignment with the simplex vertices than softmax outputs. These results highlight that logits offer a better trade-off between alignment and spread, which is a desirable property for obtaining a transferable and robust representation according to Chen et al. (2022); Wang & Isola (2020). The impact of this property is evident in the experiments reported in Sec. 4. We demonstrate that similar results hold for different model architectures and datasets (see Appendix C).

### 3.3 Deterministic Angular Correction Under Expanding Class Update

In this section, we address a key challenge in achieving compatibility with hyperspherical simplex representation when new classes are introduced during model updates. The expansion of the classifier induces an angular misalignment in the prototypes of old classes, compromising compatibility. We formally characterize the angular misalignment and introduce a deterministic linear transformation that compensates for alignment across updates. Our approach leverages the geometry of the hyperspherical simplex representation of Eq. 4 and Eq. 5, requiring no learnable parameters or data forwarding, thereby providing a training-free mechanism to ensure compatibility even as the model expands to accommodate new classes.

When the model is updated to incorporate additional classes, the classifier is expanded to accommodate the new outputs. This expansion alters the geometric structure of the probability simplex, simultaneously shifting its center and inducing an angular displacement in the prototypes of previously existing classes, which depend on the center's position (Eq. 2). This displacement (as illustrated by the dashed arrows in Fig. 4) reconfigures the old prototypes to form a new, higher-dimensional simplex structure. Consequently, angular misalignment emerges between the class prototypes at step $k$ and their updated counterparts at step

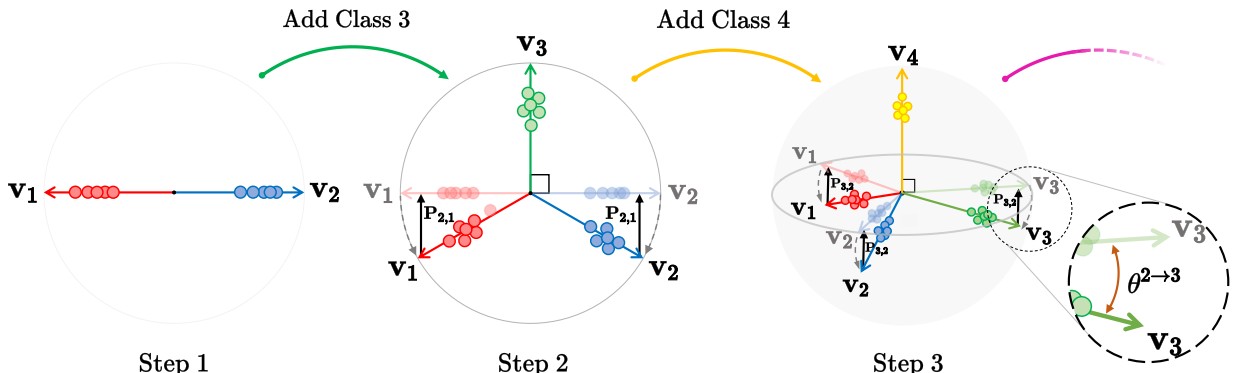

Figure 4: Deterministic linear transformation of class prototypes of Eq. 2 across model updates to which hyperspherical simplex representations of Eq. 4 and Eq. 5 align. From left to right: Step 1: 1-simplex with two classes (red and blue); Step 2: 2-simplex (new orthogonal class added in green); Step 3: 3-simplex (new orthogonal class added in yellow). Dashed arrows at steps 2 and 3 show how class prototypes rotate to form the new simplex. Black dashed circle on the right shows the angular misalignment $\theta_c^{k \to t}$ between old and new class prototype of class 3 between steps 2 and 3. Black solid arrows indicate the linear transformation $\mathbf{P}_{t,k}$ that aligns class prototypes in the new and old representations.

$t$ across model updates. This angular misalignment, derived in Appendix D, is quantified as:

$$\theta_c^{k \to t} = \arccos\left(\sqrt{\frac{C^k - 1}{C^k} \cdot \frac{C^t}{C^t - 1}}\right), \tag{6}$$

where $C^k$ and $C^t$ represent the number of classes at steps $k$ and $t$, respectively, with $C^k \leq C^t$. From Eq. 6, it follows that: (a) when $C^t = C^k$, $\theta_c^{k \to t} = 0$, the representations are therefore aligned; (b) when $C^k < C^t$, $\theta_c^{k \to t} > 0$, class prototypes across steps are not aligned.

To obtain alignment of prototypes when new classes are added, the angular misalignment of Eq. 6 between the simplex geometries is compensated with a deterministic linear transformation. This transformation is defined as $\mathbf{P}_{t,k} = \begin{bmatrix} \mathbf{V}^k | \mathbf{0} \end{bmatrix}$, where $\mathbf{0}$ is a zero matrix of dimension $C^k \times (C^t - C^k)$. It consists of two terms: (a) the matrix $\mathbf{V}^k$ acts on the first $C^k$ dimensions, mapping the new prototypes of the existing classes ($\{\mathbf{v}_c^t\}_{c=1}^{C^k}$) onto the hyperplane spanned by the old counterparts ($\{\mathbf{v}_c^k\}_{c=1}^{C^k}$). It removes the angular misalignment introduced by the new classes at step $t$, thereby aligning these prototypes across updates; (b) the matrix $\mathbf{0}$, which discards the new information in the additional dimensions ($C^t - C^k$) introduced by the newly added classes, as they are not present in the representations at step $k$. The following two propositions formalize the properties of $\mathbf{P}_{t,k}$, where comparisons across update steps are performed after embedding the lower-dimensional prototypes into the higher-dimensional space via zero-padding.

**Proposition 3.4** (Proof in Appendix E). *When updating the model from step $k$ to $t$, if the number of classes increases from $C^k$ to $C^t$, the class prototypes defined in Eq. 2 for the new classes are orthogonal to the prototypes in the previous step $k$.*

**Proposition 3.5** (Proof in Appendix F). *Let $\mathbf{v}_c^t$ be the prototype vector for class $c$ defined in Eq. 2 at step $t$, then $\mathbf{u}_c^t = \mathbf{P}_{t,k}\mathbf{v}_c^t$ is aligned with the prototype $\mathbf{v}_c^k$ at step $k$.*

From Prop. 3.4 and Prop. 3.5, it follows that our linear transformation $\mathbf{P}_{t,k}$ preserves alignment of representations also when new classes are introduced during updates. Prop. 3.4 establishes that the prototypes of the newly added classes at step $t$ are orthogonal to the hyperplane spanned by the old prototypes at step $k$. When the probability simplex is expanded to accommodate new classes, its center shifts, thereby redefining the existing class prototypes in all dimensions. As a result, the updated prototypes are no longer confined to the old hyperplane and acquire nonzero components along its orthogonal directions. This induces the angular misalignment $\theta_c^{k \to t}$ defined in Eq. 6 Prop. 3.5 demonstrates that $\mathbf{P}_{t,k}$ compensates for this angular

misalignment by removing the orthogonal component identified in Prop. 3.4. Fig. 4 shows how the hyperspherical simplex representations evolve when introducing new training classes (dashed arrows) and how $\mathbf{P}_{t,k}$ aligns class prototypes across steps (solid arrows), removing the angular displacement between old and new prototypes of the same class.

The deterministic linear transformation $\mathbf{P}_{t,k}$ does not rely on any learnable parameters or require forwarding data through a model, as it is independent of the stochasticity of the learned representation. Thus, it provides a training-free alignment of hyperspherical simplex representations (Eq. 4 and Eq. 5) across model updates, which satisfies the compatibility definition as demonstrated in the following section, even when new classes are added.

We refer to the method utilizing hyperspherical simplex representations derived from softmax outputs (Eq. 4) as Probability Simplex Projection (PSP), and to the method utilizing logits (Eq. 5) as Logits Simplex Projection (LSP). Pseudocode for gallery indexing with PSP/LSP and for comparing query-gallery representations across across model updates is provided in Appendix G.

### 3.4 PSP and LSP Verify Backward-Compatibility Definition

In the following, we theoretically demonstrate that PSP and LSP satisfy both backward-compatibility inequalities and discuss the assumptions, scope, and geometric implications of this result.

**Theorem 3.6** (***Compatibility Theorem***; Proof in Appendix A.). *Consider regular simplex representations as defined in Eq. 4 and Eq. 5 obtained at different update steps. Under the assumptions that (a) class features follow a von Mises-Fisher distribution* $\mathrm{vMF}(\boldsymbol{\mu}, \kappa)$ *and (b) the concentration $\kappa$ of class distributions increases across updates, it follows that the compatibility inequalities of Eq. 1a and Eq. 1b are satisfied in expectation.*

The $\mathrm{vMF}(\boldsymbol{\mu}, \kappa)$ distribution models hyperspherical features whose mean direction $\boldsymbol{\mu}$ represents the class prototype of Eq. 2, while $\kappa$ controls their spread around the prototype. This modeling approach has a geometric foundation introduced in Liu et al. (2016); Banerjee et al. (2005), which has provided the basis for representation learning on the hypersphere across various computer vision applications, including face recognition (Liu et al., 2017; Deng et al., 2019) and variational autoencoders (Davidson et al., 2018). Empirical support of this assumption is provided by Seo et al. (2021); Scott et al. (2021), who demonstrate that learned feature representations in the final layers of deep neural networks can be statistically modeled with a vMF distribution.

Neural scaling laws (Prato et al., 2021; Caballero et al., 2023) further show that using larger datasets or models yields backbone features (i.e., pre-classifier representations) that are increasingly concentrated around their prototypes (i.e., a larger $\kappa$). A similar concentration effect occurs in the internal representation of a model when expanding the number of classes during an update, as the margin between adjacent classes decreases due to class means arranging themselves to maximize the minimum one-vs-rest margin (Jiang et al., 2024; Yaras et al., 2022). Since, in a linear classifier, class means of backbone features and classifier weights collapse onto each other during training (Papyan et al., 2020), this margin reduction is reflected from the feature space to the logit space. Therefore, increasing the concentration of backbone features leads to increasing the concentration of logits and their corresponding softmax outputs—due to the monotonic property of the softmax function Lapin et al. (2016); Yang & Koyejo (2020). In Appendix H, we provide empirical evidence of this phenomenon on the CIFAR-10 dataset.

Proof in Appendix A shows that (a) Eq. 1a holds in expectation if prototypes of the same class are aligned across updates and (b) Eq. 1b holds in expectation when the angle between any two prototypes of different classes is greater than $\pi/2$. PSP and LSP representations inherently satisfy these two conditions due to their geometric properties. The first condition is satisfied because prototypes leverage the one-hot encoded labels as a fixed reference; when new classes are introduced during updates, this alignment is preserved through the deterministic linear transformation $\mathbf{P}_{t,k}$ as demonstrated in Prop. 3.5. The second condition is satisfied because the class prototypes form a regular simplex configuration, where, by definition, the angle between any two distinct prototypes is greater than $\pi/2$ (Pernici et al., 2019b; Papyan et al., 2020; Pernici et al., 2021a).

**Discussion.** Theorem 3.6 provides a conditional guarantee whose strength depends on the concentration of the learned representations. The separation in expected cosine distance between same-class and different-class pairs is governed by the mean resultant length of the vMF distribution (see Eq. 13 in Appendix A), which is monotonically increasing in $\kappa$ (Hornik & Grün, 2014). When representations are poorly concentrated (small $\kappa$), mean resultant length approaches zero and the expected cosine distances for both same-class and different-class pairs converge toward 1, reducing the separation required to satisfy Eq. 1b. The guarantee therefore degrades continuously and analytically as concentration decreases. Importantly, Theorem 3.6 does not require high model accuracy, but rather that $\kappa$ of the updated model at step $t$ exceeds that of the old model at step $k$. The NC1 curves of Fig. 3 and Appendix C further confirm that concentration increases consistently throughout training toward convergence.

More generally, Theorem 3.6 is not exclusive to PSP and LSP. Under the assumptions of a vMF feature distribution and increasing concentration $\kappa$ of class distributions across model updates, it extends to any regular simplex representation whose geometry remains aligned across updates, including the one learned according to fixed $d$-simplex classifiers such as those considered by Biondi et al. (2023a; 2024) (see Remark 1). The main motivation for PSP and LSP, however, is that they satisfy these conditions without any modification to the network architecture, in particular without requiring the classifier head to be fixed a priori.

This broader scope also clarifies the relation between Theorem 3.6 and the findings of Biondi et al. (2024). Their results are derived by modeling representations as uniformly distributed hyperballs in Euclidean space. In that setting, Euclidean distances between features are unbounded, and inter-class separation depends jointly on the displacement of class centers and the radii of the hyperballs. Consequently, their conclusions are specific to Euclidean hyperball geometry and lead to the result that Eq. 1a is satisfied when feature clusters of the same class remain aligned across model updates, whereas Eq. 1b is not. In contrast, we study representations normalized onto the unit hypersphere—a standard setting in retrieval tasks—where inter-class separation is governed solely by the angular relations between class prototypes, independently of their magnitudes. This hyperspherical formulation further allows us to model features as following a von Mises-Fisher distribution, thereby enabling the derivation of a closed-form analytical framework (Lemma A.1) for verifying the backward-compatibility constraints in expectation. By contrast, in Biondi et al. (2024), the two constraints were evaluated empirically through Monte Carlo simulation, without yielding an exact analytical solution.

Finally, the geometric simplex structure underlying Theorem 3.6 should also be distinguished from the Simplex ETF condition observed in Neural Collapse (Papyan et al., 2020). In NC, the Simplex ETF is a geometrical configuration in which internal features converge during the terminal phase of training. However, while the Simplex ETF defines the relative geometry between class features (angles and norms), it does not define their absolute position in the representation space: two models can each converge to a Simplex ETF while having the same class prototypes with different directions. Therefore, Simplex ETF does not yield representation alignment across model updates. In the representations of Eq. 4 and Eq. 5, the class prototypes $\mathbf{v}_c^k = \mathbf{e}_c - \mathbf{o}^k$ define a specific instance of a regular simplex whose coordinates are determined by the canonical basis vectors $\mathbf{e}_c$. It provides: (i) implicit alignment of same-class prototypes across model updates; (ii) exact quantification of the misalignment introduced by new classes via Eq. 6; and (iii) deterministic correction of that misalignment via $\mathbf{P}_{t,k}$, as proven in Prop. 3.5.

### 3.5 Dimensionality Reduction via Top-$k$ Sparsification

Hyperspherical simplex representations—from softmax outputs (Eq. 4 for PSP) or from logits (Eq. 5 for LSP)—have the feature dimensionality equal to the number of classes. Although increasing the number of classes to train a model yields higher-quality and more transferable feature representations (Ramanujan et al., 2023; Hong et al., 2024), it has a limitation when using hyperspherical simplex representations in large-scale retrieval scenarios: the memory footprint for gallery storage scales linearly with both the number of gallery samples and the class count, making practical application computationally prohibitive. To address this, we exploit the monotonic and top-$k$ calibration properties of the softmax function (Lapin et al., 2016; Yang & Koyejo, 2020), which provide theoretical justification for retaining only the most informative components of the features while discarding less relevant ones. This yields sparse representations that maintain discriminative power while significantly reducing memory requirements. The top-$k$ sparsification strategy is

therefore well-suited for PSP and LSP representations, as both are intrinsically derived from the softmax function, providing a theoretically motivated feature dimensionality reduction.

Following the theoretical analysis of Lapin et al. (2016), the softmax function intrinsically preserves the ranking of class relevance: The Bayes-optimal solution under softmax cross-entropy loss constitutes a monotonic reparameterization of the class posterior probabilities, thus preserving their natural ordering. Specifically, the Bayes-optimal classifier satisfies (Lapin et al., 2016):

$$f_y^*(x) = \begin{cases} \log p_y(x) + \log \alpha & \text{if } p_y(x) > 0 \\ -\infty & \text{otherwise,} \end{cases} \tag{7}$$

where $\alpha > 0$ represents a positive scaling constant and $p_y(x) = P(Y = y \mid X = x)$ denotes the true posterior probability that the label is $y$ given input instance $x$. Eq. 7 establishes that optimal predicted scores are monotonically increasing functions of the true class probabilities, since the logarithm preserves order for positive arguments. This means that softmax outputs maintain top-$k$ calibration properties. Similarly, since the softmax function satisfies the strong rank-preserving condition (Yang & Koyejo, 2020), i.e., $\mathbf{z}_i > \mathbf{z}_j \iff \sigma(\mathbf{z})_i > \sigma(\mathbf{z})_j$, the ranking is preserved also in the logits.

Based on this, we adopt a top-$k$ sparsification strategy for both PSP and LSP by retaining only the top-$k$ entries of the feature vector, while setting the remaining entries to zero. This addresses the scalability considerations shared by other methods (Biondi et al., 2023a; Pernici et al., 2021b; Yang et al., 2022; 2023; Biondi et al., 2024; 2023b) that utilize fixed simplex-based feature representations across model updates. Crucially, by producing highly sparse vectors, it also enables the use of efficient approximate nearest neighbor search libraries (e.g., PySparNN), which accelerates large-scale retrieval. In Sec. 4, we provide empirical evidence demonstrating that top-$k$ sparsification effectively enables scalable feature representations for both PSP and LSP.

## 4 Experimental Results

In this section, we present a comprehensive empirical evaluation to assess the backward-compatibility of our PSP and LSP methods across various update scenarios. We first detail the evaluation protocol and metrics in Sec. 4.1. In Sec. 4.2, we benchmark our methods against state-of-the-art approaches under established compatibility settings to ensure a fair comparison. In Sec. 4.3, we introduce a novel experimental setting to demonstrate the inherent compatibility of independently pretrained models that are downloaded from public repositories.

### 4.1 Evaluation Setting

Following prior work on model compatibility (Shen et al., 2020; Zhang et al., 2022; Meng et al., 2021; Jaeckle et al., 2023), we adopt image retrieval as the prototypical application, where the goal is to retrieve images from a gallery $\mathcal{G} = \{(\mathbf{x}_i, y_i)\}_{i=1}^{N_g}$ using queries from a query set $\mathcal{Q} = \{\mathbf{x}_i\}_{i=1}^{N_q}$. Since verifying compatibility as defined in Eq. 1a and Eq. 1b is intractable at large scale (Shen et al., 2020)—requiring evaluation of every pairwise distance between samples—we follow standard practice in the literature and adopt the Empirical Compatibility Criterion proposed by Shen et al. (2020):

$$M(\phi^t, \phi^k; \mathcal{Q}, \mathcal{G}) > M(\phi^k, \phi^k; \mathcal{Q}, \mathcal{G}) \quad \text{with } t > k, \tag{8}$$

where $M$ represents a performance metric. The term $M(\phi^t, \phi^k; \mathcal{Q}, \mathcal{G})$ denotes the *cross-model accuracy* which is the performance value obtained when query features are extracted from the updated model $\phi^t$ and gallery features from the previous model $\phi^k$. Conversely, $M(\phi^k, \phi^k; \mathcal{Q}, \mathcal{G})$ represents the *same-model accuracy* where both gallery and query features are extracted using the same model $\phi^k$. Similar to previous works (Ramanujan et al., 2022; Zhou et al., 2023; Ricci et al., 2024; Jaeckle et al., 2023; Ricci et al., 2025), the Cumulative Matching Characteristic (CMC@1) is used as the retrieval performance metric $M$ and the full test set of each dataset is used as both the query set and gallery set. The CMC@1 metric evaluates the retrieval accuracy by calculating pairwise cosine distances between query and gallery feature representations,

Table 1: Comparative analysis of compatibility performance under the "Extended Classes" benchmark on CIFAR-100 for multiple updates. Dark and light blue indicate respectively the highest and the second-highest value for each metric.

| | 2 steps | | | 5 steps | | | 20 steps | | | 50 steps | | |
|---|---|---|---|---|---|---|---|---|---|---|---|---|
| METHOD | $AC$ | $AA$ | $ACA$ | $AC$ | $AA$ | $ACA$ | $AC$ | $AA$ | $ACA$ | $AC$ | $AA$ | $ACA$ |
| Baseline | 0 | 29.63 | 0 | 0 | 13.60 | 0 | 0 | 04.22 | 0 | 0 | 02.29 | 0 |
| BCT | 1 | 40.64 | 35.51 | 0 | 31.03 | 0 | 0 | 22.66 | 0 | 0.01 | 15.78 | 0.84 |
| RACT | 1 | 44.19 | 45.09 | 0.60 | 30.43 | 16.53 | 0.05 | 21.61 | 2.42 | 0 | 16.71 | 0 |
| AdvBCT | 0 | 35.32 | 0 | 0.40 | 26.13 | 11.67 | 0.02 | 19.79 | 0.19 | 0 | 14.70 | 0.03 |
| LCE | 1 | 43.48 | 40.71 | 0.10 | 32.63 | 04.63 | 0 | 20.95 | 0.25 | 0 | 13.81 | 0.03 |
| CoReS | 1 | 40.94 | 35.21 | 0.30 | 30.65 | 12.29 | 0.12 | 24.88 | 4.61 | 0.11 | 23.50 | 3.91 |
| PSP | 1 | 36.31 | 29.05 | 0.90 | 26.04 | 21.76 | 0.52 | 20.56 | 12.99 | 0.39 | 19.42 | 10.76 |
| LSP | 1 | 41.14 | 36.38 | 0.70 | 30.36 | 21.91 | 0.44 | 24.11 | 13.26 | 0.36 | 22.68 | 11.25 |

excluding the query image from the gallery set to avoid a trivial match. A retrieval operation is considered successful when the nearest gallery sample in the feature space belong to the same class label as the query sample.

To evaluate compatibility across multiple model updates, we construct a compatibility matrix following Biondi et al. (2023a;b); Bui et al. (2025):

$$
\mathbf{C}_{t,k} = \begin{cases} 0 & \text{if } t < k \\ M\big(\phi^k, \phi^k; \mathcal{Q}, \mathcal{G}\big) & \text{if } t = k \\ M\big(\phi^t, \phi^k; \mathcal{Q}, \mathcal{G}\big) & \text{if } t > k \end{cases}
\tag{9}
$$

This matrix structure enables a comprehensive analysis of model interactions in all the update combinations, extending beyond the two or three training steps evaluated in previous studies (Shen et al., 2020; Zhou et al., 2023; Zhang et al., 2021). Compatibility evaluation is performed by comparing entries of the compatibility matrix according to the Empirical Compatibility Criterion of Eq. 8. To provide a comprehensive assessment and obtain metrics that capture the overall behavior of the methods across multiple updates, we adopt the metrics defined in Biondi et al. (2023a), which summarize the compatibility matrix values into distinct measures, conceptually analogous to the ones employed in continual learning by Lopez-Paz & Ranzato (2017):

- *Average Compatibility (AC)*, which quantifies how frequently compatibility is achieved across $T$ model updates: $AC = \frac{2}{T(T-1)} \sum_{t=2}^{T} \sum_{k=1}^{t-1} \mathbb{1}\big(\mathbf{C}_{t,k} > \mathbf{C}_{k,k}\big)$, where $\mathbb{1}$ denotes the indicator function.

- *Average Accuracy (AA)*, which representing the average retrieval performance obtained over $T$ model updates, regardless of compatibility: $AA = \frac{2}{T(T+1)} \sum_{t=1}^{T} \sum_{k=1}^{t} \mathbf{C}_{t,k}$.

In addition to these, we quantify the trade-off between retrieval accuracy and compatibility using the following metric:

- *Average Compatibility Accuracy (ACA)*, represents the mean retrieval performance when compatibility is achieved: $ACA = \frac{2}{T(T-1)} \sum_{t=2}^{T} \sum_{k=1}^{t-1} \mathbf{C}_{t,k}$ s.t. $\mathbf{C}_{t,k} > \mathbf{C}_{k,k}$.

## 4.2 Standard Compatibility Benchmarks

In this section, we compare PSP and LSP with state-of-the-art methods in standard compatibility benchmarks. Following standard practice (Shen et al., 2020; Zhou et al., 2023; Meng et al., 2021; Biondi et al., 2023a; Pan et al., 2023), we evaluate performance in three distinct scenarios: (a) extended classes, where

Table 2: Comparative analysis of compatibility performance under the "Extended Classes" benchmark on ImageNet-1K and Google Landmarks v2 for multiple updates. Dark and light blue indicate respectively the highest and the second-highest value for each metric.

| | ImageNet-1K | | | | | | Google Landmark v2 | | | | | |
| | 2 steps | | | 5 steps | | | 2 steps | | | 5 steps | | |
| METHOD | $AC$ | $AA$ | $ACA$ | $AC$ | $AA$ | $ACA$ | $AC$ | $AA$ | $ACA$ | $AC$ | $AA$ | $ACA$ |
|---|---|---|---|---|---|---|---|---|---|---|---|---|
| Baseline | 0 | 37.52 | 0 | 0 | 16.18 | 0 | 0 | 07.66 | 0 | 0 | 07.87 | 0 |
| BCT | 1 | 48.81 | 45.86 | 0.20 | 34.96 | 5.28 | 1 | 09.37 | 08.22 | 0 | 08.46 | 0 |
| RACT | 1 | 55.17 | 51.16 | 0.10 | 40.94 | 6.20 | 0 | 11.01 | 0 | 0 | 09.99 | 0 |
| AdvBCT | 0 | 53.07 | 0 | 0 | 43.91 | 0 | 1 | 11.12 | 09.67 | 0 | 10.73 | 0 |
| LCE | 0 | 47.22 | 0 | 0 | 32.67 | 0 | 1 | 09.41 | 08.03 | 0.40 | 10.31 | 3.80 |
| CoReS | 0 | 46.11 | 0 | 0 | 37.67 | 0 | – | – | – | – | – | – |
| PSP | 1 | 49.10 | 39.80 | 1 | 35.20 | 31.09 | 1 | 08.74 | 08.24 | 0.90 | 08.53 | 7.21 |
| ↪ w top-256 | 1 | 49.11 | 39.81 | 1 | 35.20 | 31.09 | 1 | 08.74 | 08.24 | 0.90 | 08.53 | 7.21 |
| LSP | 1 | 50.72 | 45.62 | 0.50 | 38.03 | 14.28 | 1 | 09.31 | 08.36 | 0.80 | 09.36 | 7.34 |
| ↪ w top-256 | 1 | 50.34 | 45.84 | 0.70 | 37.96 | 24.85 | 1 | 09.30 | 08.35 | 0.80 | 09.34 | 7.31 |

updates introduce new classes; (b) extended backbone, where the model architecture is upgraded; and (c) a combination of both.

According to Shen et al. (2020) and compatibility literature, in all these experimental scenarios, unless otherwise specified, the new model is trained from scratch (randomly initialized), which represents the most challenging setting for achieving compatibility: it prevents the model from leveraging optimization biases to learn compatible representation starting directly from the previous model (an initial compatible state). To learn the representations, classification is used as a surrogate task. Implementation details of the standard compatibility benchmark experiments are presented in Appendix I.

Our evaluation includes a comprehensive comparison against leading methods: BCT (Shen et al., 2020), RACT (Zhang et al., 2021), AdvBCT (Pan et al., 2023), LCE (Meng et al., 2021), and CoReS (Biondi et al., 2023a). For reproducibility, we utilize the official public implementations for all compared methods. All experiments were performed on two Nvidia A100 40GB GPUs.

### 4.2.1 Extended Classes Benchmark

The experiments are conducted as follows: each dataset is partitioned into $T$ sequential tasks, each introducing a distinct and equal number of novel classes. At each training step $t$, the model is trained from scratch on the cumulative data from all tasks encountered so far (i.e., tasks 1 to $t$). Following Zhou et al. (2023); Ricci et al. (2024); Jaeckle et al. (2023); Zhou et al. (2022), the model is evaluated on image retrieval over the entire test set at each training step $t$, creating a semi-open-set evaluation scenario where the model must retrieve images from all classes (including those not yet encountered during training) while being trained incrementally on an expanding subset of classes from tasks 1 to $t$.

Tab. 1 reports compatibility results according to the Average Compatibility, Average Accuracy, and Average Compatibility Accuracy on CIFAR-100 (Krizhevsky, 2009) with a ResNet-18 backbone. The same metrics are shown in Tab. 2 for ImageNet-1K (with ResNet-50 as backbone) and Google Landmarks v2 (Weyand et al., 2020) (with ResNet-18 as backbone).

Overall, Tab.1 and Tab.2 demonstrate that PSP and LSP achieve competitive performance with respect to the performance of compared methods in both Average Compatibility ($AC$) and Average Compatibility Accuracy ($ACA$), while maintaining high Average Accuracy ($AA$). The performance gap between our proposed method and other approaches becomes increasingly pronounced as the number of sequential tasks increases, with compatibility metrics showing progressively larger advantages when evaluated on 50 tasks (CIFAR-100) and 5 tasks (ImageNet-1K). This phenomenon can be attributed to the fundamental limitation of regularization- and mapping-based approaches: during each model update, the new representation is aligned exclusively

with the immediately preceding one, rather than being simultaneously aligned with representations from all old models. This sequential alignment strategy allows misalignment in representation to accumulate progressively over time, leading to lower values of compatibility. In contrast, PSP and LSP rely on the fixed geometry of a regular simplex and are therefore not subject to this. Among the compared approaches, CoReS is the only other method that employs a similar simplex-based structure. Notably, CoReS demonstrates high average accuracy ($AA$) values comparable to LSP. However, its reliance on fixed pre-allocated class prototypes can lead to representation misalignment due to the influence of unused prototypes during loss optimization (Fang et al., 2021). These unused pre-allocated prototypes generate negative gradients that interfere with the representation learning for actual classes, thus inducing misalignment with respect to their specific fixed class prototypes. Furthermore, the method suffers from scalability issue. The memory requirements, driven by both the pre-allocated fixed classifier and the additional linear layer needed to adapt the model architecture to the classifier, grows substantially with the number of classes. For large-scale dataset (e.g., in Google Landmark v2 of Tab. 2), this memory demand makes training computationally infeasible. Conversely, in the less challenging scenario involving only a single model update, PSP and LSP exhibit no significant performance difference compared to other methods, with the exception of RACT and AdvBCT, which are specifically designed to enhance compatible model performance in this particular case.

Notably, for all number of steps in both Tab. 1 and Tab. 2, LSP yields higher $AA$ but slightly lower $AC$ to PSP, reflecting the logits' higher spread and lower alignment to the simplex geometry with respect to softmax outputs. Consequently, LSP often achieves higher $ACA$ than PSP. To visually assess the retrieval performance of the 5-step update on CIFAR-100 across all model updates, the corresponding compatibility matrices are reported in Figure 9 in Appendix J.

The rows labeled "$\rightarrow$ w top-256" in Tab. 2 report PSP and LSP compatibility performance using the top-$k$ sparsification, as discussed in Sec. 3.5. Selecting the top-256 values increases $AC$ and $ACA$ for both PSP and LSP, while leaving $AA$ almost unchanged. In the large-scale Google Landmark v2 dataset, where the PSP and LSP representation dimensionality reaches more than 80k in the last model update, top-$k$ sparsification achieves substantial memory savings (smaller memory occupation for each gallery sample thanks to a sparse coding of feature vectors) without any performance degradation and also preserves model compatibility across updates.

Table 3: Comparison of PSP vs. LSP across different top-$k$ values for $AC$, $AA$, and $ACA$ on 5 steps on ImageNet-1K under the "extended classes" benchmark of Tab. 1

| Top-$k$ with $k =$ | PSP | | | LSP | | |
|---|---|---|---|---|---|---|
| | $AC$ | $AA$ | $ACA$ | $AC$ | $AA$ | $ACA$ |
| 256 | 1 | 35.20 | 31.09 | 0.7 | 37.96 | 24.85 |
| 384 | 1 | 35.20 | 31.09 | 0.7 | 38.00 | 24.87 |
| 512 | 1 | 35.20 | 31.09 | 0.5 | 38.03 | 14.24 |
| 768 | 1 | 35.20 | 31.09 | 0.5 | 38.01 | 14.28 |
| 896 | 1 | 35.20 | 31.09 | 0.5 | 38.03 | 14.28 |
| 1000 | 1 | 35.20 | 31.09 | 0.5 | 38.03 | 14.28 |

dates. Tab. 3 confirms that these gains persist across a broad range of $k$; especially for PSP, compatibility metrics remain the same, confirming the theory of Lapin et al. (2016) and the scalability of our proposed approaches.

### 4.2.2 Extended Backbone Benchmark

In this experimental setting, we maintain the same dataset across all training steps (with no introduction of additional class categories during model updates) while increasing model architectural complexity across 2 steps (i.e., ResNet-18 $\rightarrow$ ResNet-50). The goal is to achieve compatibility when model updates are performed by changing the model architecture, rather than introducing of novel training classes. Results for CIFAR-100 and ImageNet-1K are summarized in Tab. 4. The majority of evaluated approaches achieve compatibility ($AC = 1$), indicating that model updates utilizing more expressive model architectures induce substantially less representation drift compared to scenarios involving new class introduction. Consequently, performance distinctions manifest primarily in Average Accuracy ($AA$) and Average Compatibility Accuracy ($ACA$) metrics. LSP matches or exceeds the strongest baselines across all evaluated datasets, achieving the highest $AA/ACA$ on CIFAR-100 and maintaining competitive performance on ImageNet-1K (within 0.1–0.2 percentage points of CoReS and RACT). These results suggest that even when model capacity changes substantially, the simplex-based alignment used by LSP preserves both compatibility and discriminative power at least as well as the compared methods. Notably, PSP and LSP attain comparable performance even

Table 4: Comparative analysis of compatibility performance under the "Extended Backbone" and "Extended Classes and Backbone" benchmarks on CIFAR-100 and ImageNet-1K. Dark and light blue indicate respectively the highest and the second-highest value for each metric.

| | Ext. Backbone | | | | | | Ext. Classes & Backbone | | | | | |
| | CIFAR-100 | | | ImageNet-1K | | | CIFAR-100 | | | ImageNet-1K | | |
| METHOD | AC | AA | ACA | AC | AA | ACA | AC | AA | ACA | AC | AA | ACA |
|---|---|---|---|---|---|---|---|---|---|---|---|---|
| Baseline | 0 | 35.13 | 0 | 0 | 40.58 | 0 | 0 | 29.88 | 0 | 0 | 35.88 | 0 |
| BCT | 1 | 53.57 | 54.06 | 1 | 61.80 | 61.43 | 1 | 43.61 | 40.79 | 1 | 50.64 | 42.98 |
| RACT | 1 | 53.80 | 54.62 | 1 | 62.15 | 62.07 | 1 | 43.66 | 41.01 | 1 | 52.74 | 48.92 |
| AdvBCT | 0 | 49.56 | 0 | 1 | 59.22 | 57.73 | 1 | 39.06 | 35.05 | 1 | 50.37 | 45.42 |
| LCE | 1 | 52.14 | 52.26 | 1 | 61.22 | 61.40 | 1 | 42.42 | 39.72 | 1 | 52.28 | 48.06 |
| CoReS | 1 | 56.67 | 57.43 | 1 | 62.03 | 63.19 | 1 | 42.61 | 37.20 | 1 | 49.10 | 39.90 |
| PSP | 1 | 54.57 | 56.01 | 1 | 68.35 | 65.28 | 1 | 38.65 | 32.84 | 1 | 48.42 | 41.24 |
| LSP | 1 | 56.76 | 57.96 | 1 | 62.00 | 63.16 | 1 | 43.08 | 37.89 | 1 | 52.13 | 48.26 |

Table 5: Compatibility between pretrained models on ImageNet-1K and Places365: (*a*) AlexNet → ResNet-50 (R-50) → RegNetX_3.2GF (RegNet) → ResNet-152 (R-152) → MaxViT_T and (*b*) ViT-B-32 → ViT-B-16 → ViT-L-16. Models are pretrained on ImageNet-1K and downloaded from Marcel & Rodriguez (2010). Dark blue numbers indicate the highest values, while light blue the second-highest values.

| | AlexNet → R-50 → RegNet → R-152 → MaxViT | | | | | | ViT-B-32 → ViT-B-16 → ViT-L-16 | | | | | |
| | ImageNet-1K | | | Places365 | | | ImageNet-1K | | | Places365 | | |
| METHOD | AC | AA | ACA | AC | AA | ACA | AC | AA | ACA | AC | AA | ACA |
|---|---|---|---|---|---|---|---|---|---|---|---|---|
| Baseline | 0 | 21.63 | 0 | 0 | 8.38 | 0 | 0 | 38.63 | 0 | 0 | 15.10 | 0 |
| PSP | 1 | 74.34 | 76.12 | 0.9 | 15.92 | 14.08 | 1 | 78.94 | 80.53 | 1 | 18.31 | 17.60 |
| LSP | 0.7 | 60.68 | 37.61 | 0.2 | 21.36 | 4.06 | 0.67 | 71.82 | 45.91 | 0.33 | 25.71 | 7.99 |

without necessitating specialized regularization, domain-specific training protocols, learnable mappings, or modifications to the model architecture.

### 4.2.3 Extended Classes And Backbone Benchmark

This experimental configuration, consisting of 2 update steps in total, integrates the two previously examined update mechanisms: at each step, the model is retrained from scratch on an increased training set and with a backbone of higher expressiveness. The results for this combined setting are presented in Tab. 4 for both CIFAR-100 and ImageNet-1K datasets.

The baseline method, which implements no compatibility preservation mechanism, consistently fails to satisfy the compatibility criterion ($AC = 0$) across all evaluated datasets, mirroring the performance observed in previous experimental configurations. Conversely, all alternative methodologies achieve satisfactory compatibility scores ($AC = 1$), demonstrating the effectiveness of their respective regularization techniques, mapping strategies, and feature space pre-allocation approaches in maintaining competitive performance across all compatibility metrics. Across both ImageNet-1K and CIFAR-100 benchmarks, our LSP method demonstrates robustness under these challenging conditions, consistently matching or closely approximating the performance of leading compared methods across both datasets, while requiring no additional computational overhead or specific training procedures. Instead, PSP exhibits marginally reduced performance compared to LSP, attributable to less spread in its representation that adversely affects its performance in the semi-open set evaluation setting, consistent with findings reported in Sec 4.2.1.

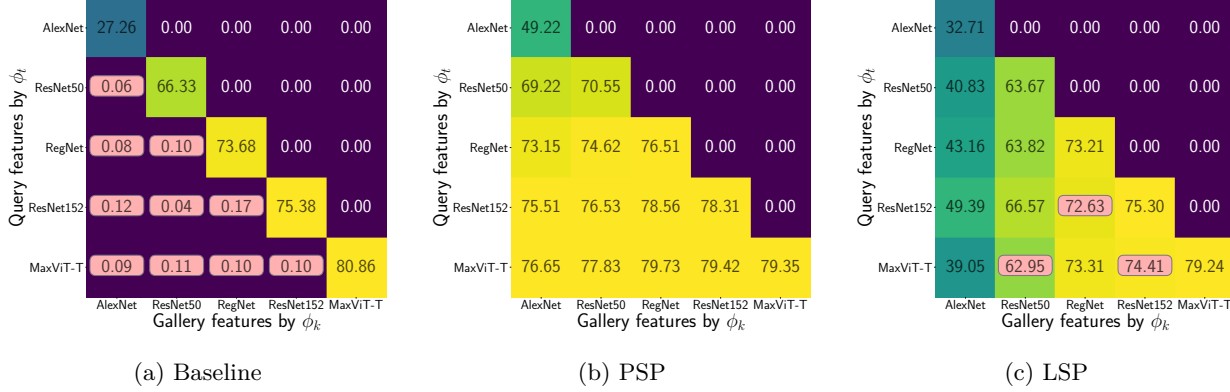

Figure 5: Compatibility between independently trained models, all pretrained on ImageNet-1K. Compatibility matrices for five steps with increasingly expressive models (AlexNet, ResNet-50, RegNetX_3.2GF, ResNet-152, and MaxViT_T). The matrices display retrieval performance on the ImageNet-1K test set. Entries that do not satisfy the compatibility criterion in Eq. 8 are highlighted with a light-red background. Model representations from: (a) encoder output (Baseline); (b) softmax outputs (PSP); (c) logits (LSP).

## 4.3 Compatibility Between Independently Trained Models From Public Repositories

Both PSP and LSP utilize hyperspherical simplex representations whose alignment across model updates is obtained with a fixed reference (one-hot label encoding) and the deterministic linear transformation $\mathbf{P}_{t,k}$, without using any additional learning procedures or external data. To demonstrate that our approach establishes backward-compatibility between independently trained models, we introduce a new benchmark where compatibility is evaluated directly (without any additional training) across two sequences of pretrained models from the publicly available torchvision repository (Marcel & Rodriguez, 2010): (a) AlexNet (Krizhevsky et al., 2012), ResNet-50 (He et al., 2016), RegNetX_3.2GF (Radosavovic et al., 2020), ResNet-152 (He et al., 2016), and MaxViT_T (Tu et al., 2022), and (b) ViT-B-32, ViT-B-16, ViT-L-16 (Dosovitskiy et al., 2021). All the models are pretrained on the ImageNet-1K dataset, and exhibit different architectures and model expressiveness. We compare PSP and LSP against a baseline that uses features obtained by $\ell_2$-normalizing the encoder outputs—the standard approach in the retrieval literature. Since all previously proposed backward-compatibility methods require additional training, adaptation procedures, or modifications to the model architecture, they cannot be compared within this benchmark.

The results, presented in Tab. 5, show that both PSP and LSP successfully maintain compatibility across all model updates. When comparing our methods, we observe a trade-off between closed-set and open-set performance. Evaluating on the ImageNet-1K dataset (closed-set condition)—the same dataset of the pretraining of the models—PSP achieves higher accuracy and compatibility ($AC$ and $AA$) compared to LSP. This improvement can be attributed to high alignment of class prototypes and limited spread of PSP feature distributions. Such properties offer significant benefits when there is no domain shift between training and test distributions, analogous to conventional classification task settings. Evaluating on the Places365 dataset (open-set condition), LSP exhibits superior overall accuracy ($AA$) compared to PSP. This behavior stems from the higher spread of LSP representation than PSP one (as shown in Sec. 3.2.2), which constitutes a more robust and transferable representation (Wang & Isola, 2020). Notably, the retrieval performance values on the main diagonal (same-model accuracy) of the compatibility matrices of Fig. 5 are nearly the same for LSP and the baseline, which instead always fails to achieve compatibility on ImageNet-1K and Places365, as shown in Tab. 5. These results confirm that using features derived from softmax output or logits do not provide performance degradation compared to using features extracted directly from encoder outputs (as in the baseline approach), yielding compatible and equivalent-performing representations for retrieval tasks.

In Appendix K, complementary compatibility matrices are presented for more detailed and visual analysis of the open-set scenario and ViT architecture updates.

## 5    Conclusions

In this paper, we propose a novel approach for compatible learning using hyperspherical simplex representations derived from softmax outputs (PSP) and logits (LSP). By leveraging the inherent alignment of these features to a specific simplex structure, our method achieves compatibility between model representations across updates without using any additional loss functions, mapping functions, or modifications to the model architecture. Consequently, PSP and LSP have the unique property of providing compatibility between models trained independently of each other, without requiring additional training or adaptation procedures. Although these representations may experience misalignment when the model is updated with novel training classes, the simplex configuration of PSP and LSP enables us to define a deterministic linear transformation that maintains alignment across model updates.

We demonstrate that these hyperspherical simplex representations satisfy both inequalities of the formal definition of backward-compatibility in expectation, thereby providing a rigorous theoretical foundation for our approach. Beyond this theoretical contribution, we show that our method yields competitive retrieval performance on standard benchmarks, outperforming existing methods in multi-update scenarios, and achieves compatibility between independently trained models from a public repository. Furthermore, we provide a theoretically grounded top-$k$ sparsification technique to mitigate the linear dimensionality scaling of PSP and LSP. Empirical results confirm that this sparsification significantly reduces storage costs without compromising retrieval performance or backward-compatibility, making it well-suited for large-scale retrieval scenarios.

**Limitations.** The geometric properties of PSP and LSP features rely on the optimization process. Specifically, the alignment of these features to a regular simplex structure is determined by the optimization dynamics of softmax cross-entropy loss functions when trained with one-hot encoded target labels. Consequently, our approach requires models to be trained using classification as a surrogate task. This requirement limits the direct applicability of our approach to models trained with alternative objectives, such as self-supervised pretraining without explicit classification heads or regression-based formulations. However, this constraint can be mitigated by fine-tuning a linear classification head following a linear-probe setting on a pre-trained encoder using a modest amount of external labeled data (see Appendix K). The computational overhead of this fine-tuning procedure is relatively minimal, as only the final classification layer requires optimization while the encoder weights can remain frozen.

A second limitation arises from the assumption of temporal consistency in the semantic structure of the data. Our method assumes the mapping between a class label and its numerical index remains constant across model updates. Changes in class ordering or label definitions would disrupt the geometric correspondence upon which the backward-compatibility of our hyperspherical simplex representations relies. However, in practical deployment scenarios, datasets exhibit different versioning dynamics compared to model architectures or training procedures. While models may undergo frequent architectural modifications and training with different paradigms, dataset evolution typically follows more conservative patterns. Modern versioning practices involve the augmentation of existing classes with additional samples or the incremental introduction of new categories, according to Lin et al. (2022).

**Statement of Broader Impact.** While backward-compatible updates can reduce the cost of re-indexing, they may also preserve existing biases in legacy galleries, and the compatibility mechanism itself could introduce or amplify disparities if performance changes unevenly across subpopulations. Evaluating these effects and understanding when gallery refresh is required (e.g., for corrective debiasing updates) are important directions for future work.

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

# A  PSP and LSP Verify Compatibility Definition

**Lemma A.1.** *Let $\mathbf{X}_1, \mathbf{X}_2 \in \mathbb{R}^d$ be independent random vectors in the $(d-1)$-hypersphere, each following a von Mises-Fisher distribution $\mathrm{vMF}(\boldsymbol{\mu}, \kappa)$. Then the expected cosine distance between $\mathbf{X}_1$ and $\mathbf{X}_2$ depends on the concentrations $\kappa_1$, $\kappa_2$ and the mean directions $\boldsymbol{\mu}_1, \boldsymbol{\mu}_2$ of the two distributions as follows:*

$$\mathbb{E}\left[1 - \mathbf{X}_1^\top \mathbf{X}_2\right] = 1 - m_d(\kappa_1)\, m_d(\kappa_2)\, \boldsymbol{\mu}_1^\top \boldsymbol{\mu}_2.$$

*Proof.* Let $\mathbf{X}_1, \mathbf{X}_2$ be two independent random vectors from two vMF distributions with parameters respectively $(\boldsymbol{\mu}_1, \kappa_1)$ and $(\boldsymbol{\mu}_2, \kappa_2)$, where $\boldsymbol{\mu_1}$, $\boldsymbol{\mu_2}$ are unit vectors indicating the mean directions around which the data are concentrated and $\kappa_1$, $\kappa_2$ are non-negative scalars that determine the concentrations of the distributions around their mean (large values of $\kappa$ correspond to high concentration). The expected angle between $\mathbf{X}_1, \mathbf{X}_2$ is given by the product of their expectations:

$$\mathbb{E}\left[\mathbf{X}_1^\top \mathbf{X}_2\right] = \mathbb{E}\left[\mathbf{X}_1\right]^\top \mathbb{E}\left[\mathbf{X}_2\right] \tag{10}$$

where the expected values of $\mathbf{X}_1$ and $\mathbf{X}_2$ are respectively (Banerjee et al., 2005):

$$\mathbb{E}\left[\mathbf{X}_1\right] = m_d\left(\kappa_1\right) \boldsymbol{\mu}_1, \tag{11}$$

$$\mathbb{E}\left[\mathbf{X}_2\right] = m_d\left(\kappa_2\right) \boldsymbol{\mu}_2, \tag{12}$$

being $m_d(\kappa)$ the expected value of the cosine of the angle between the random vector and $\boldsymbol{\mu}$, which is expressed as:

$$m_d(\kappa) = \frac{I_{\frac{d}{2}}(\kappa)}{I_{\frac{d}{2}-1}(\kappa)}$$

with $I_\nu$ the modified Bessel function of the first kind of order $\nu$.

From Eq. 10, Eq. 11, and Eq. 12, we obtain:

$$\mathbb{E}\left[\mathbf{X}_1^\top \mathbf{X}_2\right] = \left(m_d\left(\kappa_1\right) \boldsymbol{\mu}_1\right)^\top \left(m_d\left(\kappa_2\right) \boldsymbol{\mu}_2\right) = m_d\left(\kappa_1\right) m_d\left(\kappa_2\right) \boldsymbol{\mu}_1^\top \boldsymbol{\mu}_2.$$

where $\boldsymbol{\mu}_1^\top \boldsymbol{\mu}_2$ is the cosine of the angle between mean directions $\boldsymbol{\mu}_1$ and $\boldsymbol{\mu}_2$.

Since the cosine distance between $\mathbf{X}_1$ and $\mathbf{X}_2$ is $(1 - \mathbf{X}_1^\top \mathbf{X}_2)$, it follows that:

$$\mathbb{E}\left[1 - \mathbf{X}_1^\top \mathbf{X}_2\right] = 1 - \mathbb{E}\left[\mathbf{X}_1^\top \mathbf{X}_2\right] = 1 - m_d(\kappa_1)\, m_d(\kappa_2)\, \boldsymbol{\mu}_1^\top \boldsymbol{\mu}_2. \tag{13}$$

which provides an analytical form that relates the expected distance between two random vectors to the parameters of the vMF distribution. $\square$

**Theorem** (***Compatibility Theorem***). *Consider regular simplex representations as defined in Eq. 4 and Eq. 5 obtained at different update steps. Under the assumptions that (a) class features follow a von Mises-Fisher distribution $\mathrm{vMF}(\boldsymbol{\mu}, \kappa)$ and (b) the concentration $\kappa$ of class distributions increases across updates, it follows that the compatibility inequalities of Eq. 1a and Eq. 1b are satisfied in expectation.*

*Proof.* Let's now consider each feature of the model represented as an independent random vector. Then the compatibility inequalities of Eq. 1a and Eq. 1b in expectation can be expressed as:

$$\mathbb{E}\left[1 - (\mathbf{X}_1^t)^\top \mathbf{X}_2^k\right] \leq \mathbb{E}\left[1 - (\mathbf{X}_1^k)^\top \mathbf{X}_2^k\right] \qquad \text{for the case of same class} \tag{14a}$$

$$\mathbb{E}\left[1 - (\mathbf{X}_1^t)^\top \mathbf{X}_2^k\right] \geq \mathbb{E}\left[1 - (\mathbf{X}_1^k)^\top \mathbf{X}_2^k\right] \qquad \text{for the case of different classes} \tag{14b}$$

where $\mathbf{X}_1, \mathbf{X}_2 \in \mathbb{R}^d$ are two independent random vectors and $\mathbb{E}\left[1 - (\mathbf{X}_1)^\top \mathbf{X}_2\right]$ their expected cosine distance and $t$ and $k$ are two distinct update steps.

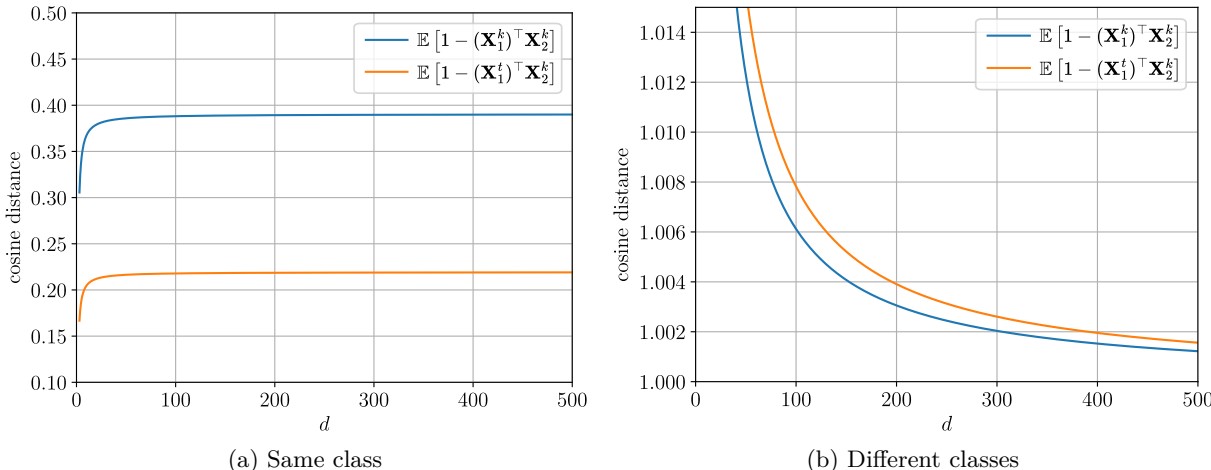

(a) Same class

(b) Different classes

Figure 6: Plot of the expected cosine distances of hyperspherical simplex representations for (a) same-class (Eq. 14a) and (b) different-class (Eq. 14b) comparisons, calculated using the closed-form solutions in Eq. 16a and Eq. 16b. The results demonstrate that the compatibility inequalities are satisfied in expectation for every value of the feature space dimension $d$.

Assuming that features follow a von Mises–Fisher distribution, according to Lemma A.1 the expected cosine distance between two independent random vectors $\mathbf{X}_1, \mathbf{X}_2 \in \mathbb{R}^d$ has the form:

$$\mathbb{E}\left[1 - \mathbf{X}_1^\top \mathbf{X}_2\right] = 1 - m_d(\kappa_1)\, m_d(\kappa_2)\, \boldsymbol{\mu}_1^\top \boldsymbol{\mu}_2 \tag{15}$$

where $\boldsymbol{\mu}_1, \boldsymbol{\mu}_2$ are the mean directions of the two distributions and $\kappa_1,\ \kappa_2$ their concentrations. Using Eq. 15, it follows that Eq. 14a and Eq. 14b become:

$$1 - m_d(\kappa_1^t)\, m_d(\kappa_2^k)\, (\boldsymbol{\mu}_1^k)^\top \boldsymbol{\mu}_2^t \le 1 - m_d(\kappa_1^k) m_d(\kappa_2^k)\, (\boldsymbol{\mu}_1^k)^\top \boldsymbol{\mu}_2^k \quad \text{for the case of same class} \tag{16a}$$

$$1 - m_d(\kappa_1^t)\, m_d(\kappa_2^k)\, (\boldsymbol{\mu}_1^k)^\top \boldsymbol{\mu}_2^t \ge 1 - m_d(\kappa_1^k) m_d(\kappa_2^k)\, (\boldsymbol{\mu}_1^k)^\top \boldsymbol{\mu}_2^k \quad \text{for the case of different classes} \tag{16b}$$

Then, given that parameter $m_d(\kappa)$ is a strictly increasing function of the concentration parameter $\kappa$ (Banerjee et al., 2005), and concentration $m_d(\kappa^t) > m_d(\kappa^k)$ across updates, it follows: $m_d(\kappa_1^t)\, m_d(\kappa_2^k) > m_d(\kappa_1^k)\, m_d(\kappa_2^k)$.

It results that:

- Eq. 16a is maximally satisfied when the mean directions of the same class remain the same across updates, i.e., $(\boldsymbol{\mu}_1^t)^\top \boldsymbol{\mu}_2^k = 1$. This is in accordance with the result obtained in Biondi et al. (2024).

- Eq. 16b is satisfied when the angle between prototypes of different classes is greater than $\pi/2$, i.e., $(\boldsymbol{\mu}_1^k)^\top \boldsymbol{\mu}_2^k = (\boldsymbol{\mu}_1^t)^\top \boldsymbol{\mu}_2^k < 0$—note that $(\boldsymbol{\mu}_1^k)^\top \boldsymbol{\mu}_2^k = (\boldsymbol{\mu}_1^t)^\top \boldsymbol{\mu}_2^k$ due to the alignment of the prototypes required to satisfy Eq. 16a.

When using PSP of Eq. 4 and LSP of Eq. 5: prototypes of the same class remain inherently aligned across model updates, even when new classes are introduced through the deterministic linear transformation $\mathbf{P}_{t,k}$ (Prop. 3.5), thus always satisfying Eq. 16a. Moreover, pairwise angles between prototypes of different classes are always the same and greater than $\pi/2$, as they are arranged in a regular simplex. This implies that Eq. 16b is satisfied. It results that the compatibility inequalities of Eq. 1a and Eq. 1b are always satisfied in expectation. □

**Remark 1** (*Extension to aligned regular simplex representations*)**.** Under the same assumptions of Theorem 3.6, the compatibility result extends to any regular simplex representation whose geometry remains aligned across model updates. In particular, this includes representations learned according to $d$-simplex fixed classifiers such as those considered by Biondi et al. (2023a; 2024).

Fig. 6 shows the analytic expression of Eq. 14a and Eq. 14b across various dimensions of hyperspherical simplex representations learned during two subsequent phases. These phases are identified by the time step $k$ to refer to the old model and $t$ to refer to the updated model which has vMF distributions with higher concentration than the old ones. Fig. 6a shows that, in the case of the same class, the value of $\mathbb{E}\left[1 - (\mathbf{X}_1^t)^\top \mathbf{X}_2^k\right]$ remains consistently lower than $\mathbb{E}\left[1 - (\mathbf{X}_1^k)^\top \mathbf{X}_2^k\right]$, which on average satisfies the condition of Eq. 1a. Fig. 6b shows that, in the case of a different class, the value of $\mathbb{E}\left[1 - (\mathbf{X}_1^t)^\top \mathbf{X}_2^k\right]$ remains consistently higher than $\mathbb{E}\left[1 - (\mathbf{X}_1^k)^\top \mathbf{X}_2^k\right]$, which on average satisfies the condition of Eq. 1b.

## B    Proof for Proposition 3.3

**Proposition** (**3.3**). *During training, while softmax outputs converge to the vertices of the probability simplex, the corresponding logits vectors align to directions as defined in Eq. 2.*

*Proof.* Given a generic sample at step $k$ of class $c$, with $1 \leq c \leq C^k$, let's consider the softmax outputs $\sigma(\mathbf{z})$ of the logits vector $\mathbf{z}$ and the one-hot encoded vector $\mathbf{c}$ with value 1 in the $c$-th position. The gradient of the softmax cross-entropy loss with respect to the logits vector $\mathbf{z}$ is:

$$\frac{\partial \mathcal{L}}{\partial z_j} = \sigma(\mathbf{z})_j - c_j = \begin{cases} \sigma(\mathbf{z})_c - 1 & \text{if } j = c \\ \sigma(\mathbf{z})_j & \text{if } j \neq c \end{cases}$$

During training, gradient descent updates the logits vector $\mathbf{z}$ as:

$$z_j \leftarrow z_j - \eta \frac{\partial \mathcal{L}}{\partial z_j} = \begin{cases} z_j - \eta(\sigma(\mathbf{z})_c - 1) & \text{if } j = c \\ z_j - \eta\, \sigma(\mathbf{z})_j & \text{if } j \neq c \end{cases}$$

Following Mixon et al. (2022) and Fang et al. (2021), we can consider logits vectors as initially distributed at the origin of the axes. Consequently, the corresponding softmax outputs is an uniform probability distribution i.e., $\mathbf{o}^k = \left[1/C^k, ..., 1/C^k\right]$.

From the above, a gradient step update results into:

$$z_j = \begin{cases} z_c = 0 - \eta\left(\frac{1}{C^k} - 1\right) = 0 + \eta\left(1 - \frac{1}{C^k}\right) & \text{if } j = c \\ z_j = 0 - \eta\left(\frac{1}{C^k}\right) = 0 + \eta\left(-\frac{1}{C^k}\right) & \text{if } j \neq c \end{cases}$$

Hence, each update acts as a scalar multiple of a vector that increases the $c$-th component while decreasing the others.

Since $\mathbf{v}_c = \mathbf{e}_c - \mathbf{o} = \mathbf{c} - \mathbf{o}$, it results that training aligns logits to class prototypes $\mathbf{v}_c$. $\qquad\square$

## C    PSP and LSP Convergence to Simplex Configuration

In the following, we present additional experiments beyond those discussed in Fig. 3, illustrating the NC hypotheses values for softmax outputs (blue curves) and logits (orange curves) at the final stage of training for ResNet-50 and DenseNet on the CIFAR-100 and Tiny-ImageNet-200 datasets.

The results in Fig. 7 present plots that are in accordance with those shown in Fig. 3 for ResNet-18 on CIFAR-100 and Tiny-ImageNet-200, confirming that logits present higher spread of feature distributions and less alignment to the simplex geometry than softmax outputs.

## D    Mathematical derivation of Eq. 6

Let $\mathbf{v}_c^k = \mathbf{e}_c - \mathbf{o}^k \in \mathbb{R}^{C^k}$ and $\mathbf{v}_c^t = \mathbf{e}_c - \mathbf{o}^t \in \mathbb{R}^{C^t}$ with $C^k \leq C^t$ the prototypes of class $c$ at two different steps $t$ and $k$. To compare prototypes across different dimensions, we extend $\mathbf{v}_c^k$ to $\mathbb{R}^{C^t}$ by zero-padding (not affecting the norm of a vector): $\tilde{\mathbf{v}}_c^k = [\mathbf{v}_c^k \mid \mathbf{0}_{C^t - C^k}] \in \mathbb{R}^{C^t}$.

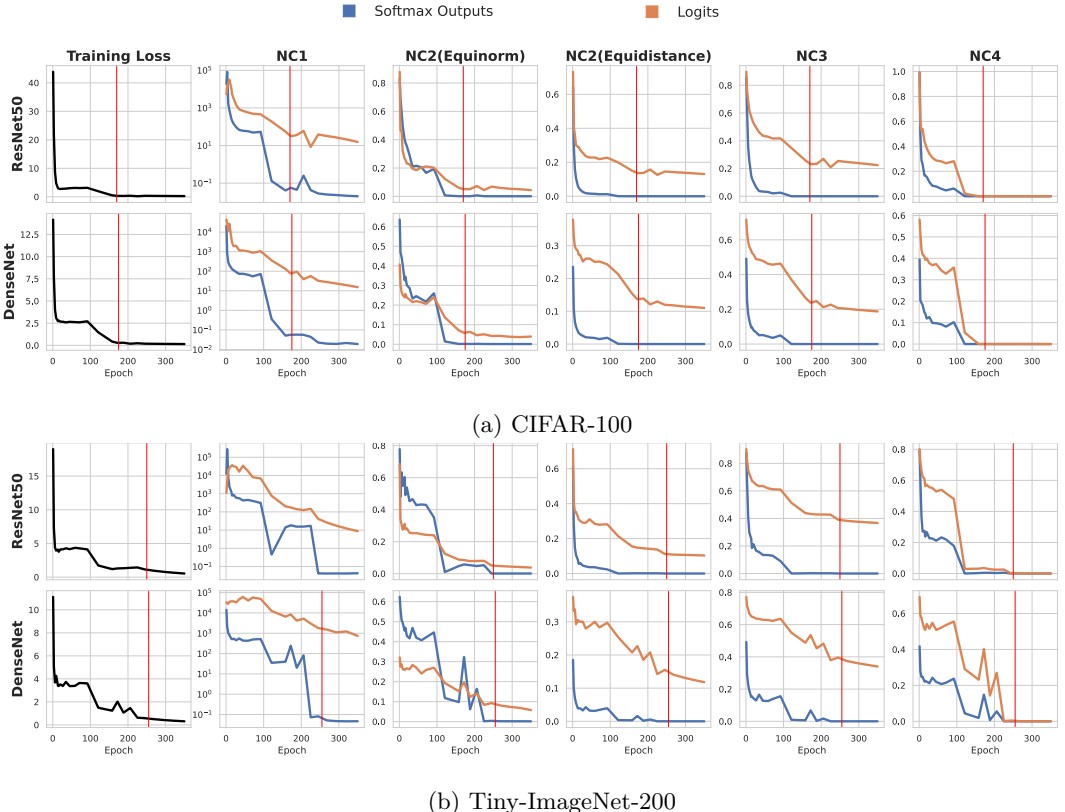

(a) CIFAR-100

(b) Tiny-ImageNet-200

Figure 7: Plots of training loss and Neural Collapse hypotheses for feature representations derived from softmax outputs (blue) and logits (orange) on CIFAR-100 and Tiny-ImageNet-200 with ResNet-50 and DenseNet. The terminal phase of training is shown with a vertical red line.

We compute the dot product between the two prototypes:

$$\tilde{\mathbf{v}}_c^{\,k} \cdot \mathbf{v}_c^t = (\mathbf{e}_c - \mathbf{o}^k) \cdot (\mathbf{e}_c - \mathbf{o}^t) = 1 - \frac{1}{C^t} - \frac{1}{C^k} + \frac{1}{C^t} = 1 - \frac{1}{C^k} = \frac{C^k - 1}{C^k}.$$

Given that the norm of the two prototypes is given by:

$$\|\tilde{\mathbf{v}}_c^{\,k}\|^2 = \frac{C^k - 1}{C^k}, \qquad \|\mathbf{v}_c^t\|^2 = \frac{C^t - 1}{C^t},$$

their cosine similarity is:

$$\cos(\tilde{\mathbf{v}}_c^{\,k}, \mathbf{v}_c^t) = \frac{\frac{C^k-1}{C^k}}{\sqrt{\frac{C^k-1}{C^k}}\sqrt{\frac{C^t-1}{C^t}}} = \sqrt{\frac{C^k - 1}{C^k} \cdot \frac{C^t}{C^t - 1}}.$$

Therefore, the misalignment angle between $\mathbf{v}_c^k$ and $\mathbf{v}_c^t$ is provided by the following formula:

$$\theta_c^{k \to t} = \arccos\left(\sqrt{\frac{C^k - 1}{C^k} \cdot \frac{C^t}{C^t - 1}}\right).$$

## E  Proof of Proposition 3.4

**Proposition** (**3.4**)**.** *When updating the model from step k to t, if the number of classes increases from $C^k$ to $C^t$, the class prototypes defined in Eq. 2 for the new classes are orthogonal to the prototypes in the previous step k.*

*Proof.* With no loss of generality, we demonstrate the proposition in the case where only one class is added between step $k$ and step $t$. For simplicity in notation, we denote $C^k = d$, resulting in $C^t = C^k + 1 = d + 1$.

From Eq. 2, a prototype vector of the $c$-th class at step $k$ is denoted as $\mathbf{v}_c^k = \mathbf{e}_c - \mathbf{o}^k$ in $\mathbb{R}^d$. Each component $v_{c,j}^k$ is:

$$v_{c,j}^k = \begin{cases} v_{c,c}^k = 1 - \frac{1}{d} & \text{if } j \text{ equals the index } c \text{ of class } y \\ v_{c,j}^k = -\frac{1}{d} & \text{otherwise} \end{cases} \tag{17}$$

Adding a new class in the training set at step $t$ adds a new vertex $\mathbf{e}_{d+1}$ in the probability simplex shifting its center to: $\mathbf{o}^t = \frac{1}{d+1} \sum_{c=1}^{d+1} \mathbf{e}_c$. The prototype vector of the new class $d+1$ is then defined as $\mathbf{v}_{d+1}^t = \mathbf{e}_{d+1} - \mathbf{o}^t$.

To measure the angle between the new prototype and the old prototypes, we compute the cosine similarity between $\mathbf{v}_{d+1}^t$ and the generic prototype $\mathbf{v}_c^k$.

According to Eq. 17, the norm of $\mathbf{v}_c^k$ and $\mathbf{v}_{d+1}^t$ are respectively:

$$\|\mathbf{v}_c^k\| = \sqrt{\sum_{j=1}^d \left(v_{c,j}^k\right)^2} = \sqrt{\sum_{\substack{j=1 \\ j \neq c}}^d \left(v_{c,j}^k\right)^2 + \left(v_{c,c}^k\right)^2} = \sqrt{\sum_{\substack{j=1 \\ j \neq c}}^d \frac{1}{d^2} + \frac{(d-1)^2}{d^2}} = \sqrt{\frac{d-1}{d^2} + \frac{(d-1)^2}{d^2}} = \sqrt{\frac{d-1}{d}}$$

and

$$\|\mathbf{v}_{d+1}^t\| = \sqrt{\sum_{j=1}^{d+1} \left(v_{d+1,j}^t\right)^2} = \sqrt{\sum_{j=1}^d \left(v_{d+1,j}^t\right)^2 + \left(v_{d+1,d+1}^t\right)^2} = \sqrt{\sum_{j=1}^d \frac{1}{(d+1)^2} + \frac{d^2}{(d+1)^2}} = \sqrt{\frac{d}{d+1}}$$

The corresponding normalized vectors are respectively:

$$\frac{\mathbf{v}_c^k}{\|\mathbf{v}_c^k\|} = \frac{\mathbf{e}_c - \mathbf{o}^k}{\sqrt{\frac{d-1}{d}}} = \begin{cases} \frac{1 - \frac{1}{d}}{\sqrt{\frac{d-1}{d}}} & \text{for class index } c \\ \frac{-\frac{1}{d}}{\sqrt{\frac{d-1}{d}}} & \text{otherwise} \end{cases} \tag{18}$$

and

$$\frac{\mathbf{v}_{d+1}^t}{\|\mathbf{v}_{d+1}^t\|} = \frac{\mathbf{e}_{d+1} - \mathbf{o}^t}{\sqrt{\frac{d}{d+1}}} = \begin{cases} \frac{1 - \frac{1}{d+1}}{\sqrt{\frac{d}{d+1}}} & \text{for class index } c \\ \frac{-\frac{1}{d+1}}{\sqrt{\frac{d}{d+1}}} & \text{otherwise} \end{cases} \tag{19}$$

In order to compute the dot product, $\mathbf{v}_c^k$ is padded with a zero component as:

$$\mathbf{v}_c^k = \left(\mathbf{e}_c - \mathbf{o}^k, 0\right)$$

which does not change its norm. Then, the dot product can be expressed as follows where the components in the $c$-th and $d+1$-th positions are put into evidence:

$$\frac{\mathbf{v}_c^k}{\|\mathbf{v}_c^k\|} \cdot \frac{\mathbf{v}_{d+1}^t}{\|\mathbf{v}_{d+1}^t\|} = \sum_{\substack{j=1 \\ j \neq c}}^d \frac{v_{c,j}^k}{\|\mathbf{v}_c^k\|} \frac{v_{d+1,j}^t}{\|\mathbf{v}_{d+1}^t\|} + \frac{v_{c,c}^k}{\|\mathbf{v}_c^k\|} \frac{v_{d+1,c}^t}{\|\mathbf{v}_{d+1}^t\|} + \frac{v_{c,d+1}^k}{\|\mathbf{v}_c^k\|} \frac{v_{c,d+1}^t}{\|\mathbf{v}_{d+1}^t\|} \tag{20}$$

Since the component $v_{c,d+1}^k = 0$, expression in Eq. 20 simplifies to:

$$\frac{\mathbf{v}_c^k}{\|\mathbf{v}_c^k\|} \cdot \frac{\mathbf{v}_{d+1}^t}{\|\mathbf{v}_{d+1}^t\|} = \sum_{\substack{j=1 \\ j \neq c}}^d \frac{v_{c,j}^k}{\|\mathbf{v}_c^k\|} \frac{v_{d+1,j}^t}{\|\mathbf{v}_{d+1}^t\|} + \frac{v_{c,c}^k}{\|\mathbf{v}_c^k\|} \frac{v_{d+1,c}^t}{\|\mathbf{v}_{d+1}^t\|}$$

Using Eq. 18 and Eq. 19 it results:

$$\frac{\mathbf{v}_c^k}{\|\mathbf{v}_c^k\|} \cdot \frac{\mathbf{v}_{d+1}^t}{\|\mathbf{v}_{d+1}^t\|} = \frac{d-1}{d(d+1)\sqrt{\frac{d-1}{d}}\sqrt{\frac{d}{d+1}}} - \frac{d-1}{d(d+1)\sqrt{\frac{d-1}{d}}\sqrt{\frac{d}{d+1}}} = 0$$

This demonstrates that a generic prototype vector $\mathbf{v}_c^k$ and a new prototype vector $\mathbf{v}_{d+1}^t$ are orthogonal. This proof can be immediately extended to model updates where more than one class are introduced. $\qquad\square$

## F   Proof of Proposition 3.5

**Proposition (3.5).** *Let $\mathbf{v}_c^t$ be the prototype vector for class c defined in Eq. 2 at step t, then $\mathbf{u}_c^t = \mathbf{P}_{t,k}\mathbf{v}_c^t$ is aligned with the prototype $\mathbf{v}_c^k$ at step k.*

*Proof.* With no loss of generality, we demonstrate the proposition in the case where only one class is added between step $k$ and step $t$. For simplicity in notation, we denote $C^k = d$, resulting in $C^t = C^k + 1 = d + 1$.

Let $\mathbf{v}_c^k = \mathbf{e}_c - \mathbf{o}^k$ and $\mathbf{v}_c^t = \mathbf{e}_c - \mathbf{o}^t$ the prototype vectors of the $c$-th class at step $k$ and $t$, respectively. Their components $v_{c,j}^k$ and $v_{c,j}^t$ are respectively defined as:

$$v_{c,j}^k = \begin{cases} v_{c,c}^k = 1 - \frac{1}{d} & \text{if } j \text{ equals the index } c \text{ of class } y \\ v_{c,j}^k = -\frac{1}{d} & \text{otherwise} \end{cases} \tag{21}$$

and

$$v_{c,j}^t = \begin{cases} v_{c,c}^t = 1 - \frac{1}{d+1} & \text{if } j \text{ equals the index } c \text{ of class } y \\ v_{c,j}^t = -\frac{1}{d+1} & \text{otherwise} \end{cases} \tag{22}$$

We consider the vector $\mathbf{u}_c^t$ in $\mathbb{R}^d$ obtained from $\mathbf{v}_c^t$ in $\mathbb{R}^{d+1}$ using the linear transformation matrix $\mathbf{P}_{t,k}$:

$$\mathbf{u}_c^t = \mathbf{P}_{t,k}\,\mathbf{v}_c^t. \tag{23}$$

where $\mathbf{P}_{t,k} = [\mathbf{V}^k \,|\, \mathbf{0}]$ with $\mathbf{V}^k = [\mathbf{v}_c^k\,]_{c=1}^d$ a $d \times d$ matrix and $\mathbf{0}$ a zero matrix $d \times 1$.

The following expressions hold for $u_{c,c}^t$ and $u_{c,j}^t$ respectively:

$$u_{c,c}^t = v_{c,c}^k v_{c,c}^t + \sum_{\substack{j=1 \\ j \neq c}}^{d} v_{c,j}^k v_{c,j}^t \tag{24}$$

$$u_{c,j}^t = v_{c,j}^k v_{c,c}^t + v_{c,c}^k v_{c,j}^t + \sum_{\substack{j=1 \\ j \neq y}}^{d-1} v_{c,j}^k v_{c,j}^t \tag{25}$$

where the $c$-th component of $\mathbf{u}_c^t$ has been put into evidence.

Substituting the expressions in Eq. 21 and Eq. 22 into Eq. 24 and Eq. 25 respectively, we obtain the following equations:

$$u_{c,c}^t = \frac{d-1}{d}v_{c,c}^t - \frac{1}{d}\sum_{\substack{j=1 \\ j \neq c}}^{d} v_{c,j}^t = \frac{d-1}{d}\frac{d}{d+1} - \frac{1}{d}(d-1)\left(-\frac{1}{d+1}\right) = \frac{d-1}{d+1} + \frac{d-1}{d(d+1)} = 1 - \frac{1}{d} = v_{c,c}^k$$

and

$$u_{c,j}^t = -\frac{1}{d}v_{c,c}^t + \frac{d-1}{d}v_{c,j}^t - \frac{1}{d}\sum_{\substack{j=1 \\ j \neq c}}^{d-1} v_{c,j}^t = -\frac{1}{d}\frac{d}{d+1} + \frac{d-1}{d}\left(-\frac{1}{d+1}\right) - \frac{1}{d}(d-2)\left(-\frac{1}{d+1}\right) = -\frac{1}{d} = v_{c,j}^k,$$

---

**Algorithm 1** Initial Gallery Indexing with PSP or LSP

---

**Require:** Model at deployment step $s$ with $C^s$ classes; gallery item $\mathbf{x}$; representation type $\texttt{repr} \in \{\texttt{PSP}, \texttt{LSP}\}$
**Ensure:** Stored gallery feature $\mathbf{h}^s(\mathbf{x})$

  1: Compute logits $\mathbf{z}^s(\mathbf{x})$
  2: **if** $\texttt{repr} = \texttt{PSP}$ **then**
  3:     Compute softmax outputs $\mathbf{p}^s(\mathbf{x}) \leftarrow \sigma(\mathbf{z}^s(\mathbf{x}))$
  4:     $\tilde{\mathbf{h}}^s(\mathbf{x}) \leftarrow \mathbf{P}_{s,s}\, \mathbf{p}^s(\mathbf{x})$
  5:     **if** $\|\tilde{\mathbf{h}}^s(\mathbf{x})\|_2 > 0$ **then**
  6:         $\mathbf{h}^s(\mathbf{x}) \leftarrow \tilde{\mathbf{h}}^s(\mathbf{x})/\|\tilde{\mathbf{h}}^s(\mathbf{x})\|_2$
  7:     **else**
  8:         $\mathbf{h}^s(\mathbf{x}) \leftarrow \mathbf{0}$
  9: **else if** $\texttt{repr} = \texttt{LSP}$ **then**
 10:     **if** $\|\mathbf{z}^s(\mathbf{x})\|_2 > 0$ **then**
 11:         $\mathbf{h}^s(\mathbf{x}) \leftarrow \mathbf{z}^s(\mathbf{x})/\|\mathbf{z}^s(\mathbf{x})\|_2$
 12:     **else**
 13:         $\mathbf{h}^s(\mathbf{x}) \leftarrow \mathbf{0}$
 14: Store $\mathbf{h}^s(\mathbf{x})$ in the gallery index

---

i.e., the components of the vector $\mathbf{u}_c^t$ are identical to those of $\mathbf{v}_c^k$.

It results that the vector obtained from $\mathbf{v}_c^t$ is aligned with the prototype vector $\mathbf{v}_c^k$ through the matrix $\mathbf{P}_{t,k}$:

$$\mathbf{u}_c^t = \mathbf{P}_{t,k}\, \mathbf{v}_c^t = \mathbf{v}_c^k.$$

This proof can be immediately extended to model updates where more than one class are introduced. □

## G  Pseudo-Code

This appendix describes how PSP and LSP are integrated into a retrieval system. Importantly, PSP and LSP are two alternative feature definitions, not post-hoc transformations to be attached to arbitrary embeddings after deployment. Whether to use features derived from softmax outputs (PSP, Eq. 4) or logits (LSP, Eq. 5) must therefore be chosen before the first deployment of the retrieval system and then used consistently to build the gallery index and process future queries. Accordingly, gallery items are indexed once using the chosen representation and stored in the retrieval database. The deterministic transformation $\mathbf{P}_{t,k}$ is only required when comparing representations extracted at different update steps, e.g., when the gallery was indexed with a model at step $k$ and an incoming query is processed by an updated model at step $t$. In the backward-compatible setting, the gallery is not re-indexed. Instead, only the query-side representation produced at step $t$ is aligned with the gallery feature space at step $k$ through $\mathbf{P}_{t,k}$. This is the inference-time protocol used in our compatibility experiments.

Alg. 1 describes the initial gallery indexing procedure with PSP or LSP. Alg. 2 describes the query-time compatibility correction used only when features extracted at step $t$ must be compared against a gallery indexed at step $k$.

Alg. 1 is used to define and store gallery features when the retrieval system is first deployed. If, after a model update, the gallery is fully re-indexed, the same algorithm is simply applied again to all gallery items using the updated model and the same chosen representation type. By contrast, Alg. 2 is used only in the backward-compatible retrieval setting, where the gallery remains fixed at step $k$ and only the incoming queries are processed by the updated model at step $t$. In this case, PSP/LSP are not re-applied to the stored gallery features; only the query-side representation is corrected through $\mathbf{P}_{t,k}$ to enable comparison with the gallery without re-indexing.

We emphasize that the deterministic transformation $\mathbf{P}_{t,k}$ is not part of the generic feature-definition step of the retrieval system. It is only required when features extracted at different update steps with increased

---

**Algorithm 2** Query-Side Compatibility Correction Across Model Updates

---

**Require:** Gallery indexed at step $k$; updated model at step $t$ with $C^t \geq C^k$; query $\mathbf{x}$; representation type
    `repr` $\in \{\texttt{PSP}, \texttt{LSP}\}$
**Ensure:** Query feature aligned with the gallery feature space at step $k$
  1: Compute logits $\mathbf{z}^t(\mathbf{x})$
  2: **if** `repr` $=$ `PSP` **then**
  3:     Compute softmax outputs $\mathbf{p}^t(\mathbf{x}) \leftarrow \sigma(\mathbf{z}^t(\mathbf{x}))$
  4:     $\tilde{\mathbf{h}}^{t \to k}(\mathbf{x}) \leftarrow \mathbf{P}_{t,k}\, \mathbf{p}^t(\mathbf{x})$
  5:     **if** $\|\tilde{\mathbf{h}}^{t \to k}(\mathbf{x})\|_2 > 0$ **then**
  6:         $\mathbf{h}^{t \to k}(\mathbf{x}) \leftarrow \hat{\mathbf{h}}^{t \to k}(\mathbf{x})/\|\tilde{\mathbf{h}}^{t \to k}(\mathbf{x})\|_2$
  7:     **else**
  8:         $\mathbf{h}^{t \to k}(\mathbf{x}) \leftarrow \mathbf{0}$
  9: **else if** `repr` $=$ `LSP` **then**
10:     $\tilde{\mathbf{h}}^{t \to k}(\mathbf{x}) \leftarrow \mathbf{P}_{t,k}\, \mathbf{z}^t(\mathbf{x})$
11:     **if** $\|\tilde{\mathbf{h}}^{t \to k}(\mathbf{x})\|_2 > 0$ **then**
12:         $\mathbf{h}^{t \to k}(\mathbf{x}) \leftarrow \tilde{\mathbf{h}}^{t \to k}(\mathbf{x})/\|\tilde{\mathbf{h}}^{t \to k}(\mathbf{x})\|_2$
13:     **else**
14:         $\mathbf{h}^{t \to k}(\mathbf{x}) \leftarrow \mathbf{0}$
15: Compare $\mathbf{h}^{t \to k}(\mathbf{x})$ against the stored gallery features $\{\mathbf{h}^k(\mathbf{g})\}_{\mathbf{g} \in \mathcal{G}}$
16: Return the ranked gallery items

---

number of classes must be compared. During training, there is no dependence on previous models, and no additional losses, alignment modules, or architectural modifications are introduced.

# H   Empirical Evidence of Concentration Increase With Model Update

In this section, we empirically validate the assumption discussed in Sec. 3.4, showing that the distributions of logits and softmax outputs become increasingly compact around their respective class prototypes as model updates progress. This behavior is analogous to an increase in the concentration parameter $\kappa$ of a vMF distribution. To quantify the compactness, we use the mean cosine distances between the feature representations and their respective class prototypes. Specifically, let $\mathcal{Z}_c = \{\mathbf{z}_i^c\}_{i=1}^{N_c}$ denote the set of extracted logits for class $c$, where $N_c$ represents the number of samples belonging to class $c$ in the dataset. The class prototype is defined as the mean of the extracted logits $\bar{\mathbf{z}}_c = \frac{1}{N_c} \sum_{i=1}^{N_c} \mathbf{z}_i^c$. The mean cosine distance between each feature and its class prototype for a class $c$ is given by $\delta_c = \frac{1}{N_c} \sum_{i=1}^{N_c} \left(1 - \frac{\mathbf{z}_i^c \cdot \bar{\mathbf{z}}_c}{\|\mathbf{z}_i^c\| \|\bar{\mathbf{z}}_c\|}\right)$. Lower values of $\delta_c$ indicate tighter clustering of logits around the class prototype, which corresponds to a higher concentration parameter $\kappa$ in a vMF distribution.

We conduct experiments on CIFAR-10 (Krizhevsky, 2009) dataset under two scenarios analogous to those in Sec. 4 to assess the concentration behavior of logits and softmax outputs across model updates.

**Extended Classes Scenario.** In this scenario, we investigate the logits concentration behavior when the number of classes is expanded during a model update. The old model is trained on the first 5 classes from CIFAR-10, and the updated model is trained on the complete set of 10 classes. Both models are a ResNet-18 architecture (He et al., 2016) trained for 100 epochs using stochastic gradient descent (SGD) with an initial learning rate of 0.1, momentum of 0.9, and a cosine annealing learning rate schedule. The results shown in Fig. 8a demonstrate that the majority of classes exhibit increased compactness in the updated model relative to the old model. This behavior aligns with a semi-open set scenario, where the new classes are unknown to the old model but are known to the updated model, thus explaining the significant gap in concentration in the last 5 classes. While a subset of classes shows marginal decreases in compactness, this behavior can be attributed to inter-class competition in the feature space when accommodating a larger number of categories. Despite this class-specific variability, the predominant trend supports the assumption of increasing concentration with class expansion during model updates.

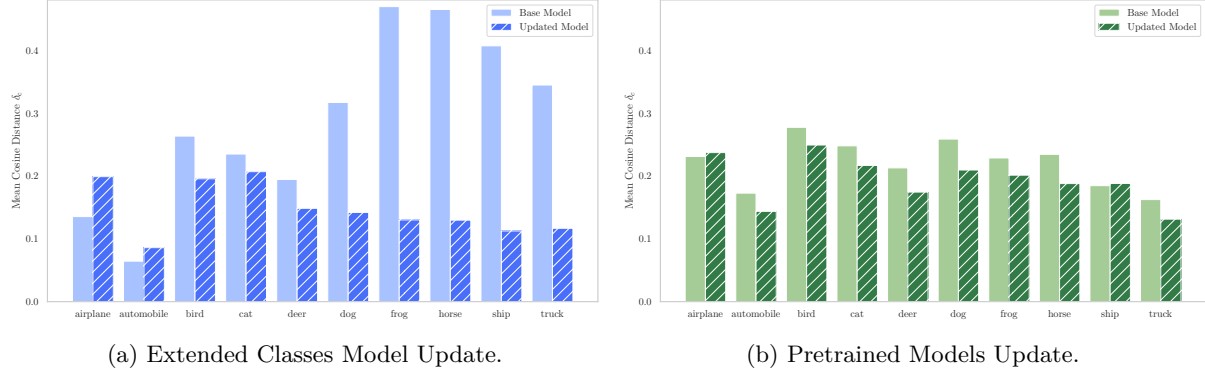

(a) Extended Classes Model Update.          (b) Pretrained Models Update.

Figure 8: Concentration analysis of class logit distributions on CIFAR-10 under two different scenarios: (a) comparison between the old model trained on 5 classes and the updated model trained on 10 classes of CIFAR-10, representing a semi-open set scenario, where the last 5 classes are unknown to the old model but known to the updated model; (b) comparison between two public released models—ResNet-18 (old model) and ResNet-50 (updated model)—pretrained on ImageNet-1K and evaluated on CIFAR-10. Lower values of $\delta_c$ indicate tighter clustering of logits around the corresponding class prototype.

**Independently Trained Models in a Public Repository Scenario.** We compare two ImageNet-1K pretrained models from Marcel & Rodriguez (2010): ResNet-18 as old model, and ResNet-50 as the updated model. We evaluate the concentration metric $\delta_c$ on the CIFAR-10 test set. As shown in Fig. 8b, the updated model yields more concentrated logit distribution than the old model, even in an open-set scenario where the evaluation distribution differs from the training one.

The empirical evidence of concentration behavior extends from the logits to the corresponding softmax outputs through the monotonic property of the softmax function (Lapin et al., 2016), demonstrating that models produce increasingly concentrated representation distribution across updates. Since the hyperspherical simplex representations of Eq. 4 and Eq. 5 are derived from the softmax outputs and logits, respectively, these empirical findings extend to both representations. This validates the assumption of an increasing concentration parameter $\kappa$ across model updates, as formalized in Theorem 3.6.

## I   Implementation Details For Standard Compatibility Benchmarks

In the following we report the implementations details we used in the experiments of Sec. 4.2.

**Experiments on CIFAR-100** (Krizhevsky, 2009). For this set of experiments the image size is $32\times32$. Both ResNet-18 and ResNet-50 architecture are used with the following hyper-parameters for training: number of epochs 120; batch size 128; SGD optimizer with learning rate that starts from 0.1 and is divided by 10 after 80 and 100 epochs; SGD momentum 0.9; weight decay $5\cdot10^{-4}$. A temperature factor of 12 is used in the cross-entropy loss to scale logits vectors during training. Training images are subjected to random cropping, horizontal flipping, and tensor normalization. For the Extended Classes benchmark, CIFAR-100 training-set is divided into chunks each having $100/T$ classes, being $T$ the training steps.

**Experiments on ImageNet-1K** (Russakovsky et al., 2015). For this set of experiments the image size is $224\times224$. Both ResNet-18 and ResNet-50 architecture are trained with the following hyper-parameters: number of epochs 90; batch size 1536; SGD optimizer with learning rate that starts from 0.1 and is divided by 10 after 30 and 60 epochs; SGD momentum 0.9; weight decay $1\cdot10^{-4}$. Training images are subjected to random cropping, horizontal flipping, color jitter, and tensor normalization, as standard for ImageNet-1K training. For the Extended Classes benchmark, ImageNet-1K training-set is divided into chunks each having $1000/T$ classes, being $T$ the training steps.

**Experiments on Google Landmarks v2** (Weyand et al., 2020). For this set of experiments the image size is $224\times224$. A ResNet-18 architecture, pretrained with ImageNet-1K, is used with the following hyper-

parameters for training: number of epochs 30; batch size 512; SGD optimizer with learning rate that starts from 0.1 and is divided by 10 after 5, 10 and 20 epochs; SGD momentum 0.9; weight decay $5 \cdot 10^{-4}$. Training images are subjected to random cropping and tensor normalization. In the initial step, we use a subset with 24393 classes. In the following steps, Google Landmark v2 training-set is divided into chunks each having $(81313 - 24393)/T$ classes, being $T$ the training steps.

## J Compatibility Matrices for 5-Step Update Scenario in the Extended Classes Benchmark

In Fig. 9, we present the compatibility matrices showing the CMC@1 retrieval metric values at each model update step for the 5-step scenario on the CIFAR-100 dataset of Tab. 1.

The compatibility matrices demonstrate that PSP and LSP not only achieve the highest compatibility scores, but also maintain superior retrieval performance throughout the update steps.

## K Compatibility Matrices of Independently Trained Models in a Public Repository Scenario

In this section, we present the compatibility matrices corresponding to the experimental results reported in Tab. 5 of Sec. 4.

Specifically, Fig. 10 shows each CMC@1 values of the compatibility matrices in the open-set scenario using Places365 as the evaluation set. The diagonal entries represent *same-model accuracy*, while the lower triangular entries show *cross-model accuracy*. These experiments involve sequential model updates using AlexNet, ResNet-50, RegNetX_3.2GF, ResNet-152, and MaxViT_T, each independently pretrained on ImageNet-1K, with Places365 serving as the test set.

Furthermore, Fig. 11 and 12 show the compatibility matrices for closed-set and open-set scenarios, respectively, using three Vision Transformer (ViT) architectures: ViT-B-32 and ViT-B-16, both trained from scratch on ImageNet-1K, and ViT-L-16, which is pretrained via self-supervised learning on ImageNet21k and subsequently fine-tuned on ImageNet-1K (Singh et al., 2022).

The results demonstrate that in the closed-set scenario, PSP achieves the highest compatibility and retrieval performance. Conversely, in open-set scenarios, while PSP maintains superior compatibility, it exhibits reduced retrieval performance. LSP achieves moderately better compatibility than the Baseline but lower than PSP; however, it substantially outperforms both Baseline and PSP in cross-model retrieval performance, as evidenced by the off-diagonal values in the compatibility matrices.

In addition to supervised ImageNet-pretrained classifiers from torchvision (Paszke et al., 2019; Marcel & Rodriguez, 2010), we extend our analysis to self-supervised pretrained encoders and vision-language pretrained models in a linear-probe setting. Specifically, we consider two families of publicly available pretrained models: (i) three checkpoints from the DINO family, namely DINO (Caron et al., 2021), DINOv2 (Oquab et al., 2023), and DINOv3 (Siméoni et al., 2025), obtained from the Hugging Face model hub (Wolf et al., 2019); and (ii) three CLIP variants pretrained by OpenAI, namely ViT-B/32, ViT-B/16, and ViT-L/14, obtained from the OpenCLIP repository (Ilharco et al., 2021). In all cases, the image encoder is used as a frozen feature extractor.

For each checkpoint, we train a bias-free linear classifier on top of the extracted features using ImageNet-1K as the training set for 30 epochs, while keeping the encoder weights frozen. We then construct PSP and LSP representations from the resulting classifier outputs, using softmax outputs and logits, respectively. As a baseline, we directly use the $\ell_2$-normalized encoder output without any additional training. Retrieval performance is evaluated on the ImageNet-1K test set, following the same protocol adopted in the main benchmark.

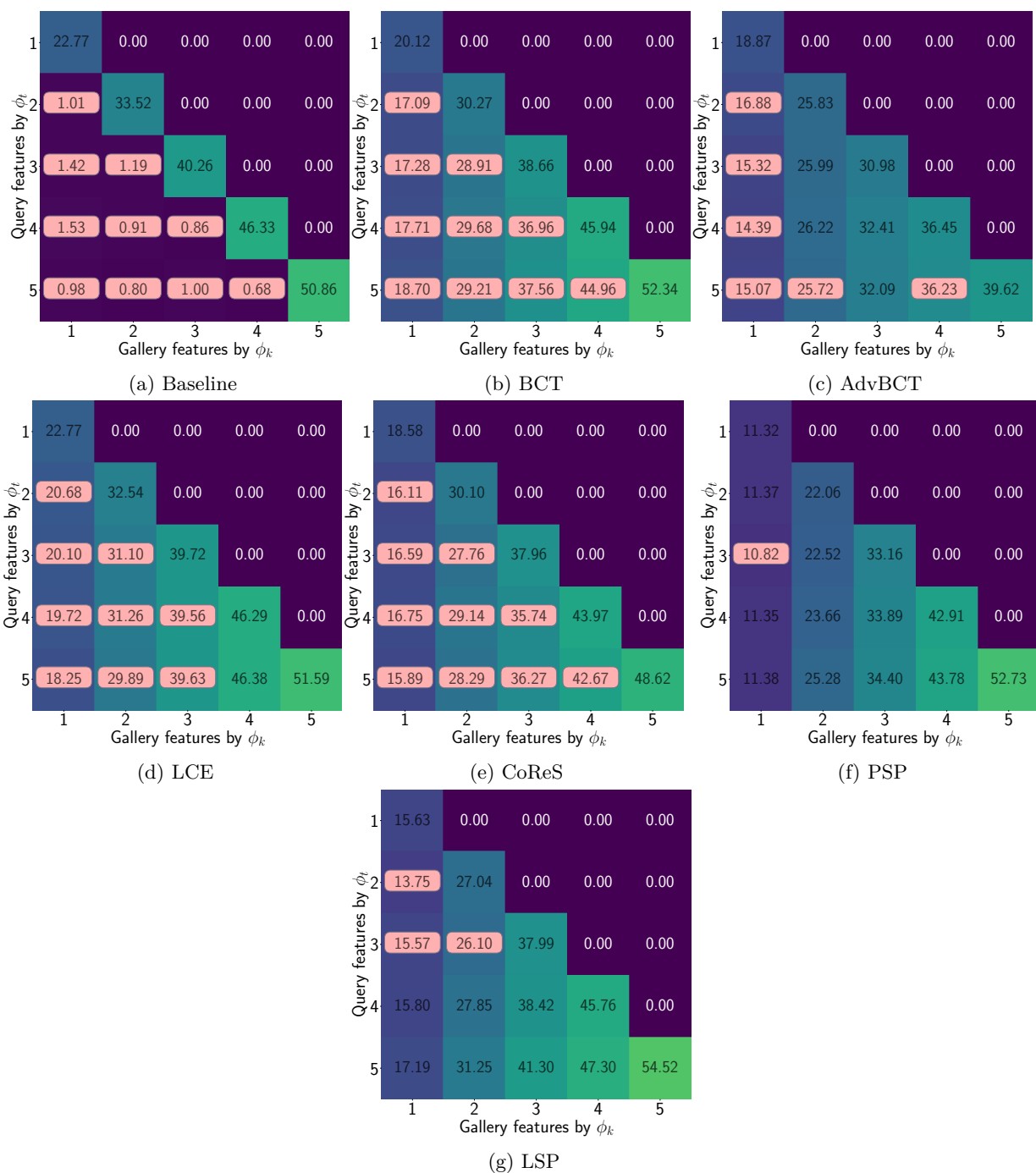

Figure 9: Compatibility matrices showing CMC@1 retrieval metric values at each model update step for the 5-step scenario on the CIFAR-100 dataset. Each matrix entry failing to satisfy the compatibility criterion defined in Eq. 8 is highlighted with a light-red background.

Tab. 6 summarizes the compatibility results obtained in this linear-probe setting on ImageNet-1K. For both DINO and CLIP model families, PSP and LSP substantially improve backward-compatibility over the baseline, confirming that the proposed framework can be extended beyond standard supervised classifiers.

For the DINO family, the observed trend is consistent with that reported for supervised ImageNet-pretrained models in Sec. 4.3. In particular, PSP achieves the strongest compatibility results, with AC = 1.0 and

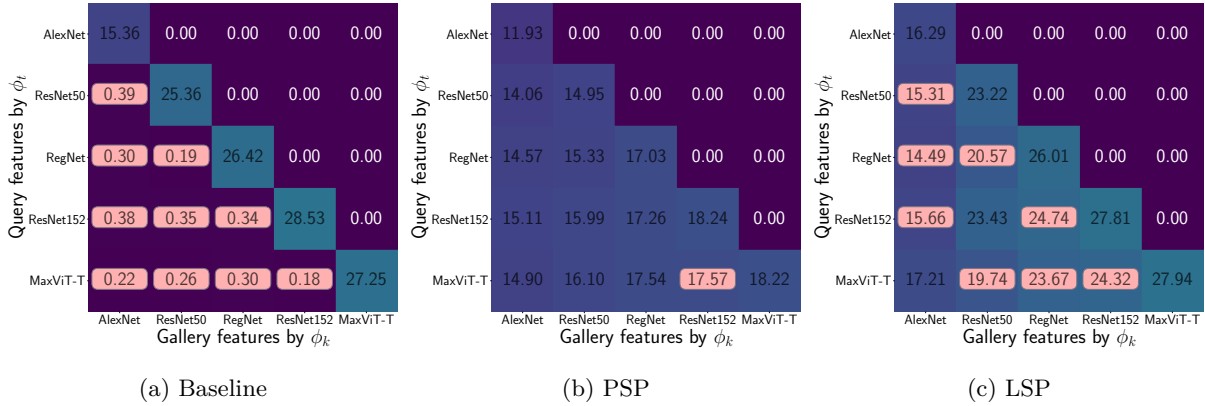

Figure 10: Open-set scenario. Compatibility matrices for five steps with increasingly expressive models (AlexNet, ResNet-50, RegNetX_3.2GF, ResNet-152, and MaxViT_T), all pretrained on ImageNet-1K. The matrices display retrieval performance on the Places365 dataset. Entries that do not satisfy the compatibility criterion in Eq. 8 are highlighted with a light-red background. Model representations obtained from: (a) encoder output (Baseline); (b) softmax outputs (PSP); (c) logits (LSP).

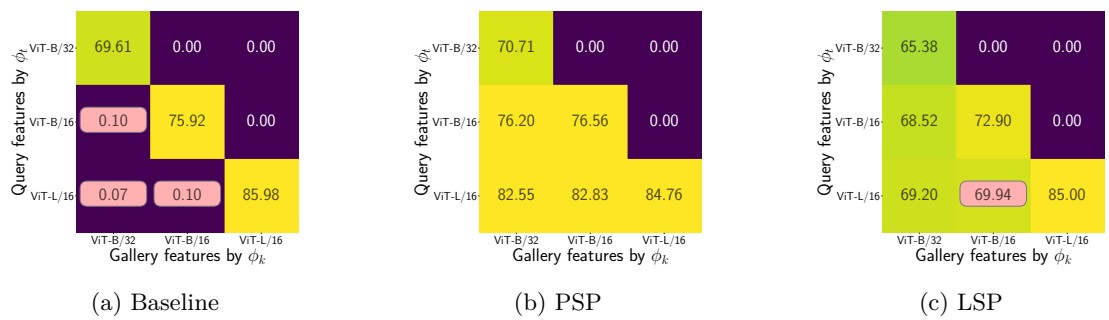

Figure 11: Closed-set scenario. Compatibility matrices for three steps with advanced Vision Transformer (ViT) model architectures (ViT-B-32, ViT-B-16, and ViT-L-16), all pretrained on ImageNet-1K. The matrices show closed-set retrieval performance on the ImageNet-1K test set. Entries that do not satisfy the compatibility criterion in Eq. 8 are highlighted with a light-red background. Model representations obtained from: (a) encoder output (Baseline); (b) softmax outputs (PSP); (c) logits (LSP).

ACA = 0.72, while LSP remains competitive with AC = 0.67 and ACA = 0.705. As shown in Fig. 13, the baseline achieves reasonable same-model retrieval performance, but cross-model retrieval remains negligible, indicating that independently pretrained self-supervised encoders do not naturally produce aligned representation spaces across updates.

For the CLIP family, we observe the same overall pattern, as shown in Fig. 14. PSP again achieves the strongest compatibility results (AC = 1.0, ACA = 0.592), while LSP improves substantially over the baseline but remains below PSP. The consistency of this behaviour across both model families provides additional empirical support for the theoretical analysis in Sec. 3.4.

Overall, these results show that a simple linear-probe protocol provides a practical way to apply PSP and LSP to pretrained encoders that do not natively expose a classification head, including both self-supervised and vision-language models.

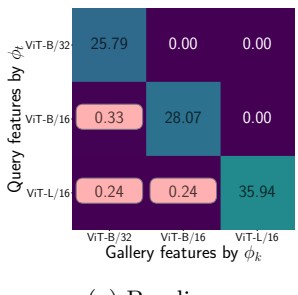 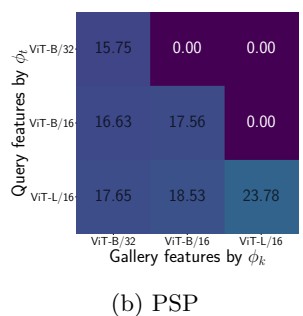 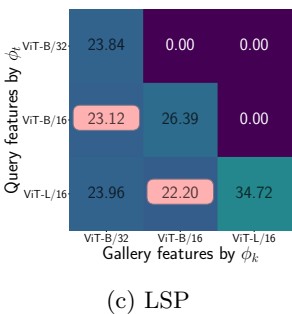

(a) Baseline      (b) PSP      (c) LSP

Figure 12: Open-set scenario. Compatibility matrices for three steps with advanced Vision Transformer (ViT) model architectures (ViT-B-32, ViT-B-16, and ViT-L-16), all pretrained on ImageNet-1K. The matrices show open-set retrieval performance on the Places365 test set. Entries that do not satisfy the compatibility criterion in Eq. 8 are highlighted with a light-red background. Model representations obtained from: (a) encoder output (Baseline); (b) softmax outputs (PSP); (c) logits (LSP).

Table 6: Compatibility results for self-supervised and vision-language pretrained encoders in a linear-probe setting on ImageNet-1K. For each checkpoint, a linear probe is trained on top of frozen backbone features. Dark blue numbers indicate the highest values, while light blue numbers indicate the second-highest values.

(a) DINO family: DINO-ViT-S $\rightarrow$ DINOv2-ViT-S/16 $\rightarrow$ DINOv3-ViT-L/16

| Method | $AC$ | $AA$ | $ACA$ |
|---|---|---|---|
| Baseline | 0 | 35.50 | 0 |
| PSP | **1.00** | **71.00** | **72.00** |
| LSP | 0.67 | 68.70 | 70.50 |

(b) CLIP family: CLIP ViT-B-32 $\rightarrow$ CLIP ViT-B-16 $\rightarrow$ CLIP ViT-L-14

| Method | $AC$ | $AA$ | $ACA$ |
|---|---|---|---|
| Baseline | 0 | 30.70 | 0 |
| PSP | **1.00** | **60.10** | **59.20** |
| LSP | 0.33 | 52.10 | 49.30 |

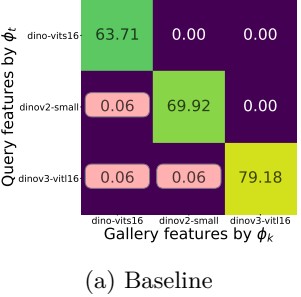 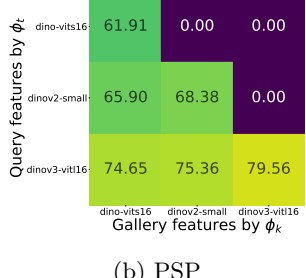 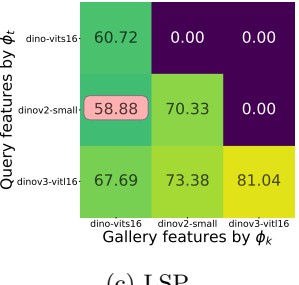

(a) Baseline      (b) PSP      (c) LSP

Figure 13: Compatibility matrices for three steps with self-supervised Vision Transformer checkpoints from the DINO family (DINO-ViT-S, DINOv2-ViT-S/16, and DINOv3-ViT-L/16) obtained from Huggingface model hub (Wolf et al., 2019). For each checkpoint, we train a linear probe on ImageNet-1K with the backbone frozen, and construct PSP/LSP features from the resulting classifier outputs. The matrices show closed-set retrieval performance on the ImageNet-1K test set. Entries that do not satisfy the compatibility criterion in Eq. 8 are highlighted with a light-red background. Model representations obtained from: (a) encoder output (Baseline); (b) softmax outputs (PSP); (c) logits (LSP).

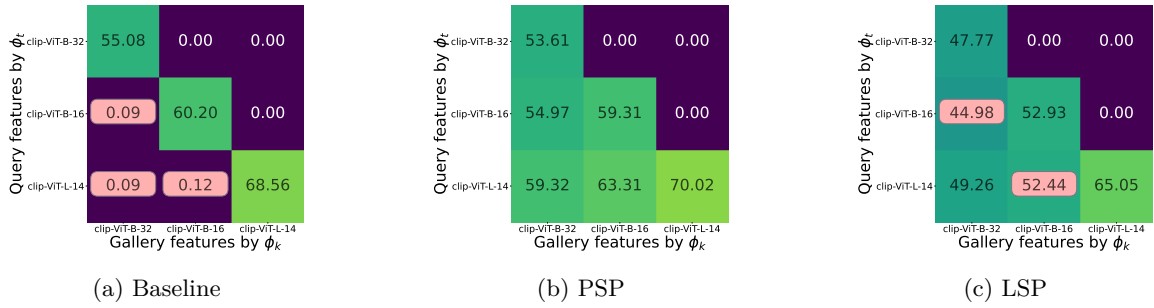

(a) Baseline        (b) PSP        (c) LSP

Figure 14: Compatibility matrices for three steps with vision-language CLIP family (ViT-B-32, ViT-B-16, and ViT-L-14) pretrained by OpenAI. For each checkpoint, we train a linear probe on ImageNet-1K with the backbone frozen, and construct PSP/LSP features from the resulting classifier outputs. The matrices show closed-set retrieval performance on the ImageNet-1K test set. Entries that do not satisfy the compatibility criterion in Eq. 8 are highlighted with a light-red background. Model representations obtained from: (a) encoder output (Baseline); (b) softmax outputs (PSP); (c) logits (LSP).

