# OpenReview forum: "Hyperspherical Simplex Representations from Softmax Outputs and Logits are Inherently Backward-Compatible"
_TMLR — Rejected by TMLR_

### Review · Reviewer_jXYC · 2026-01-14

**Summary Of Contributions:**

This paper provides a method to achieve backward compatibility in the scenario where models are continuously updated. In particular, the paper presents the PSP and the LSP method, which projects the softmax values and the logits given by the model onto the unit sphere, followed by a correction of the angular distortion. The paper justifies the approach by theoretically showing that the projected method, when facing a new class, preserves the angles between the class centers. Moreover, the paper presented experimental results on the image retrieval task on multiple datasets to demonstrate the empirical performance of the method.

**Audience:**

Yes

**Audience Explanation:**

Ensuring backward compatibility along model updates seems like an important task, so I think the paper tackles an important topic.

**Claims And Evidence:**

Yes

**Claims Explanation:**

The claims are mainly on the proposed method (PSP and LSP), the theoretical guarantee (provable backward compatibility), and the experimental performance. The paper as is covers the three major aspects.

**Requested Changes:**

1. I find the paper's writing quite confusing. In particular, although the paper provided a good description of how to process the logits or the softmax values to ensure compatibility, it does not mentioned how to integrate the proposed method in existing tasks. For instance, one does not know from the writing whether we should re-apply the method to the output of the model on datum in the gallery every single time, or do we only apply the method to the output of the model on the query. I suggest the author to add a section describing the overall algorithm of applying PSP/LSP and the angular corrections to existing tasks.

2. The statement in Section 2 that "the optimization of the cross-entropy loss leads to softmax outputs to converge toward the vertices $\mathbf{e}\_c$ lacks proper justification. As far as I know, this can only be achieved when the model can interpolate the training data or can map the data to a space that is linearly separable. I suggest the author to make the claim more rigorous.

3. In Eq. (4), the paper need to mention how to handle the case $\sigma(\mathbf{z}^k) = \mathbf{o}^k$.

4. I find the ETF condition from Neural Collapse to be an important assumption for Theorem 3.6. The paper need to make this clearer in both the statement of Theorem 3.6 and in the abstract and introduction.

---

> ### Author Response · Authors · 2026-03-13
> **Response to Reviewer jXYC (1/2)**
>
> We thank the Reviewer for recognizing the importance of our work on backward-compatible model updates and its relevance to the TMLR community. We also appreciate the acknowledgment that the manuscript's claims are well-supported through the proposed methods, theoretical analysis, and experimental evidence.
> We are encouraged that the Reviewer found our theoretical justification compelling. In particular, we are glad that both the proposed deterministic angular correction between class prototypes and the empirical validation across multiple datasets were received positively.
>
>
> In the following, we provide detailed responses to each of the requested changes and questions raised by the Reviewer.
>
> **[1. Integration of the proposed method into existing retrieval tasks]**
>
> We thank the Reviewer for pointing out this ambiguity. To address it directly: the gallery is indexed once and is not re-processed for each query.
>
> PSP and LSP define the retrieval representation itself, using either softmax outputs (PSP, Eq. 4) or logits (LSP, Eq. 5). This choice is made before the first deployment of the retrieval system, and gallery items are indexed from the beginning using the selected representation.
>
> When the model is updated, the gallery remains fixed. For each incoming query, we extract its representation with the updated model and apply the deterministic transformation $\mathcal{P}_{t,k}$, followed by $\ell_2$-normalization, only on the query side, so that it is comparable with the stored gallery features at step $k$. Therefore, in the backward-compatible setting, the method is not re-applied to all gallery items at every search step.
>
> `Action taken in the manuscript`: We revised and expanded the pseudo-code Appendix G to make this deployment protocol explicit, clarifying the distinction between initial gallery indexing with PSP/LSP and the query-side transformation used only for cross-model comparison.
>
>
> **[2. Statement on softmax outputs convergence]**
>
> We thank the Reviewer for this important observation. The original statement was indeed incomplete. The convergence of softmax outputs toward the vertices $e_c$ (equivalently, driving the cross-entropy loss toward zero) requires that the model has sufficient capacity to fit the training data. In practice, this corresponds to the learned representations being linearly separable in the penultimate feature space [Soudry et al., 2018, Fang et al., 2021]. While modern deep neural networks are typically over-parameterized and empirically operate in this regime (commonly referred to as the terminal phase of training in the Neural Collapse literature), we agree that these assumptions should be stated explicitly to ensure rigor.
>
> `Action taken in the manuscript`: We have revised the statement in Section 3.2.1 accordingly:
>
> *Original*:
> >The optimization of the cross-entropy loss leads softmax outputs to converge toward the vertices $e_c$, $\ldots$
>
> *Revised:*
> >Assuming the model has sufficient capacity to map training data to a linearly separable representation space (Soudry et al., 2018, Fang et al., 2021), the optimization of the cross-entropy loss leads softmax outputs to converge toward the vertices $e_c$, $\ldots$
>
> [Soudry et al., 2018] Soudry, D., Hoffer, E., Nacson, M. S., Gunasekar, S., & Srebro, N. (2018). The implicit bias of gradient descent on separable data. Journal of Machine Learning Research, 19(70), 1–57.
>
> [Fang et al., 2021] Cong Fang, Hangfeng He, Qi Long, and Weijie J Su. Exploring deep neural networks via layer-peeled model: Minority collapse in imbalanced training. Proceedings of the National Academy of Sciences, 2021.
>
>
> **[3. Softmax($z^k$) = $o^k$]**
>
> We thank the Reviewer for identifying this corner case. When the softmax output is exactly the uniform distribution $o^k$ (i.e., maximum uncertainty), the $\ell_2$-normalization in Eq. 4 is undefined, as the input to the normalization, after the projection $\mathbf{P}_{k,k}$, becomes the zero vector.
>
> `Action taken in the manuscript`: In the revised manuscript, we have added a footnote to Eq. 4 to address this explicitly: when $\sigma(z^k)$ = $o^k$, we define the representation as the zero vector $\mathbf{0}$, which corresponds to zero similarity with all class prototypes and is mathematically well-posed. It is worth noting that, while this case is theoretically possible, it is unlikely to occur in practice, as trained classifiers typically produce non-uniform output distributions [Hendrycks & Gimpel, 2017].
>
> [Hendrycks & Gimpel, 2017] Dan Hendrycks and Kevin Gimpel. A baseline for detecting misclassified and out-of-distribution examples in neural networks. ICLR 2017.

---

> ### Author Response · Authors · 2026-03-13
> **Response to Reviewer jXYC (2/2)**
>
> **[4. ETF condition is an important assumption for Theorem 3.6]**
>
> We thank the Reviewer for this interesting comment. The simplex geometry is implicit in the theorem statement through the explicit references to Eq. 4 and Eq. 5: these equations define representations whose prototype geometry is a specific instance of a regular simplex, as established in Sec. 3.2.
>
> We believe there may be a terminological overlap here, and we clarify below the distinction between the Simplex ETF configuration in Neural Collapse and the specific simplex geometry induced by Eq. 4 and Eq. 5 in our formulation.
> In NC, the Simplex ETF is a geometrical configuration in which internal features converge during the terminal phase of training. However, while the Simplex ETF defines the relative geometry between class features (angles and norms), it does not define their absolute position in the representation space: two models can each converge to a Simplex ETF while having the same class prototypes with different directions. Therefore, Simplex ETF does not yield representation alignment across model updates.
>
> The representations of Eq. 4 and Eq. 5 used in our approach are different. Their class prototypes $\mathbf{v}_c^k = \mathbf{e}_c - \mathbf{o}^k$ (Eq. 3) define a specific instance of a regular simplex whose coordinates are defined by the canonical basis vectors $\mathbf{e}_c$. This provides three properties that are not achievable under the Simplex ETF assumption:
> 1. implicit alignment of same-class prototypes across model updates;
> 2. exact quantification of the misalignment introduced by new classes via Eq. 6;
> 3. deterministic correction of that misalignment via $\mathbf{P}_{t,k}$, as proven in Prop. 3.5.
>
>
> `Action taken in the manuscript`: We acknowledge that making the simplex geometry of Eq. 4 and Eq. 5 more explicit in the Theorem 3.6 statement would improve readability, and we revised the manuscript accordingly. We also added a clarifying paragraph at the end of Sec. 3.4 emphasizing the difference between this geometry and the Simplex ETF one.

---

### Review · Reviewer_Mx5d · 2026-02-15

**Summary Of Contributions:**

**Summary**: Models are frequently updated as new data, better architectures or new tasks emerge. However, the case can be made that even as the model is updated, the representations remain compatible. In certain cases, the old model may not be available as a reference to maintain compatibility This paper proposes a method to align representations across independently trained models by utilizing the softmax representations, and aligning them by using the centre of the simplex to maintain the position across model updates. For the case when the number of classes increases across model updates, a projection matrix that maintains backward compatibility is used. Backward compatibility is the property that the distance between representations from the same class does not increase over models updates, while the distance between representations from different classes does not decrease. It is shown that under the assumption that the features for each class follow a von Mises-Fisher distribution with a concentration that increases across updates, the proposed method satisfies compatibility. As the number of classes increases, the paper proposes using a simple dimensionality reduction technique that simply only retains the top-$k$ entries of the logit or cross-entropy output.

The experimental evaluation focuses on the image retrieval task, with the aim that as each newly trained model is introduced, the alignment of representations does not impact the task accuracy. The experiments are conducted across 4 key settings. In the first, the number of classes at every time step is increased. In the second, the model backbone is updated (with models of increasing complexity used) while the number of classes is kept the same. In the third, both are changed at each time step. Finally, the representations from independently trained models from public repositories are aligned. The results demonstrate that while the proposed method does not perform better than all previous methods, it is computationally more efficient than competing methods.

**Strengths**:
+ The paper is well-written, with most concepts explained clearly, digestible proofs and a well-laid out experimental section.
+ The theorems although simple, seem sound as long as the assumptions around the class features and their concentration are satisfied.

**Weaknesses**:
- I find the problem the paper is tackling extremely niche. While the overall problem of backward-compatibility is an important one, the paper only focuses on representation-level compatibility. It would have benefited the paper to also discuss decision-level compatibility.
- For class-wise representations after the softmax layer to cluster along the canonical directions, the model needs to have high accuracy. The paper does not discuss the case when the model has low accuracy, and thus this assumption is not satisfied.
- The proposed method will only work to align representations at the logit or later layer in a classification task. Representations from a deeper layer that may aid retrieval cannot be aligned in the proposed method, especially as the number of classes increases. While this is acknowledged in the limitations, and a fix using linear probes is proposed, it would have benefited the paper to experimentally demonstrate this setting.
- Models are also often updated to remove the impact of certain data points (unlearning) or to make them robust to new emerging threat models (robustness). These possibilities are not considered in the paper.

**Audience:**

Yes

**Audience Explanation:**

There will be clearly members of the community interested in this work as it touches on the general issue of model updates, with connections to image retrieval tasks as well as neural feature collapse for some of the claims around the LSP method. However, my belief is that interest may be limited due to the paper's limited contributions as highlighted in the weaknesses above.

**Broader Impact Concerns:**

There is no broader impact statement, and it would be nice to have one to discuss potential bias amplification resulting from proposed methods, where corrective updates for bias correction may not work as intended.

**Claims And Evidence:**

Yes

**Claims Explanation:**

**Theoretical**: The claims in the theorems are straightforward when the stated assumptions are satisfied. I checked the proofs, but not carefully and it is possible I missed some errors.

**Empirical**: The empirical claims around *performance* are tempered, and there are several cases when the competing methods perform better. However, the claims around the improved scalability of the proposed method require better justification using empirical evidence, and actual timing computations.

**Requested Changes:**

1. Make the connection to decision-level compatibility (Critical): The proposed method directly relies on the softmax outputs for alignment, so it should be simple to also maintain decision-level compatibility.
2. Provide empirical evidence for increased scalability compared to existing methods (Critical)
3. What happens when the model has low accuracy and representations do not cluster according to canonical directions as assumed ? (Critical)
4. Discuss other situations in which models are updated (Nice to have)

---

> ### Author Response · Authors · 2026-03-13
> **Response to Reviewer Mx5d (1/5)**
>
> We thank the Reviewer for their careful reading of our manuscript and their insightful comments. We are pleased that the Reviewer appreciates the quality of the work, including the clarity of the presentation, the rigor of the theoretical analysis, and the organization of the experimental section.
>
> The Reviewer correctly identifies that, in certain practical settings, the previously trained model may not be available as a reference for enforcing backward-compatibility, which limits the applicability of existing methods. This limitation directly motivates our work: rather than relying on a previously learned model, our method exploits representations that are inherently aligned across independently trained models, up to a deterministic mapping, thereby removing this dependency altogether. As the Reviewer notes, the representation dimensionality of our approach is directly dependent on the number of classes in the model output, which can be larger than the internal representation and may increase across model updates. We propose an effective and theoretically supported dimensionality reduction technique that retains only the top-$k$ entries of the logit or softmax output.
>
> We also appreciate the Reviewer's acknowledgment that the assumptions underlying our theoretical results—such as the concentration of features around class prototypes, formalized via a von Mises-Fisher distribution with increasing concentration across updates—are well-motivated and empirically supported.
>
> Regarding the backward-compatibility property written in the Reviewer summary:
> >Backward compatibility is the property that the distance between representations from the same class does not increase over models updates, while the distance between representations from different classes does not decrease.
>
> We would like to take this opportunity to make explicit a subtle but important distinction: backward-compatibility formally operates on cross-model distances rather than intra-model clustering quality over model updates. Specifically, the definition of Shen et al. (2020) requires that the updated model's representations $\mathbf{h}^t$ remain comparable to those of the old model $\mathbf{h}^k$. Formally, for features of the same class, cross-model distances should not exceed old-model distances (Eq. 1a), and for features of different classes, cross-model distances should not fall below different-class old-model distances (Eq. 1b). This is a stronger requirement than intra-model clustering quality, as it directly ensures that a new query can be matched against an old gallery without re-indexing, which is the core practical motivation of our work and backward-compatibility literature.
>
> In the following, we address all the weaknesses and requested changes raised by the Reviewer.

---

> ### Author Response · Authors · 2026-03-13
> **Response to Reviewer Mx5d (2/5)**
>
> **[1. Connection to decision-level compatibility]**
>
> We thank the Reviewer for this insightful suggestion. Our work primarily addresses representation-level compatibility for retrieval tasks (e.g., new model queries against an old indexed gallery), which is distinct from decision-level compatibility (preserving the model's predictions across updates).
>
> Our hyperspherical simplex representations are derived directly from logits and softmax outputs—typically used for obtaining model predictions—which is why our method appears to relate to decision-level compatibility, as predictions are the argmax of softmax outputs. However, extending our approach to enforce decision-level compatibility is non-trivial, as we clarify the key distinctions below. Our method leverages the logit and softmax output spaces not to enforce per-sample predictive alignment, but to induce a globally geometric structure—hyperspherical simplex representations—that aligns the representation spaces of old and new models across updates. In contrast, decision-level compatibility pursues the distinct objective of minimizing the Negative Flip Rate (NFR), defined as the proportion of samples where the new model misclassifies instances correctly predicted by the old model (Yan et al., 2021). This requires constraining the new model's decision boundaries through sample-wise distillation techniques, such as Kullback-Leibler (KL) divergence or logit matching, rather than optimizing the geometry of the representation space.
>
> In practice, these two compatibility paradigms—representation-level and decision-level—address different objectives. Representation-level methods, such as BCT and RACT, do not appear well-suited for reducing NFR, as evidenced by empirical results in Table 1 of ELODI (Zhao et al., 2024) and Table 1 of MTP (Ricci et al., 2026). Conversely, decision-level distillation techniques fail to achieve the representational alignment necessary for cross-model retrieval, as demonstrated in prior works (Shen et al., 2020; Meng et al., 2021; Ricci et al., 2025).
>
> Their formal definitions further underscore this divergence: backward-compatible constraints (Eq. 1a and Eq. 1b in our manuscript) for representation-level compatibility, whereas decision-level compatibility typically focuses on explicit minimization of NFR. Our primary contribution remains the introduction of hyperspherical simplex representations, which theoretically satisfy both backward-compatibility constraints at the representation level.
>
> [Zhao et al., 2024] Yue Zhao, Yantao Shen, Yuanjun Xiong, Shuo Yang, Wei Xia, Zhuowen Tu, Bernt Schiele, and Stefano Soatto. Elodi: Ensemble logit difference inhibition for positive-congruent training. IEEE Trans. Pattern Anal. Mach. Intell., 46(12):7529–7541, 2024.

---

> ### Author Response · Authors · 2026-03-13
> **Response to Reviewer Mx5d (3/5)**
>
> **[2. Improved scalability of the proposed method]**
>
> We thank the Reviewer for this request. Below, we provide a scalability analysis of PSP/LSP with top-$k$ sparsification, measuring both per-query processing time and memory footprint across a wide range of feature dimensionalities.
>
> All experiments were conducted on an NVIDIA GeForce RTX 3060 (11.6 GB VRAM) with a gallery of 100,000 vectors and 5,000 queries (randomly generated), averaged over 5 independent runs. Top-$k$ sparsification used $k$ = 128.
>
> *Per-Query Processing Time (mean ± std, ms/query):*
>
> | Dim | Softmax/q | Projection/q | Top-$k$/q |
> |---:|:---:|:---:|:---:|
> | 1,024 | 0.0003 ± 0.0006 | 0.0004 ± 0.0005 | 0.0007 ± 0.0010 |
> | 4,096 | 0.0001 ± 0.0000 | 0.0014 ± 0.0000 | 0.0004 ± 0.0000 |
> | 8,192 | 0.0002 ± 0.0000 | 0.0054 ± 0.0001 | 0.0007 ± 0.0000 |
> | 16,384 | 0.0005 ± 0.0000 | 0.0208 ± 0.0001 | 0.0012 ± 0.0000 |
> | 49,152 | 0.0024 ± 0.0001 | 0.1884 ± 0.0010 | 0.0037 ± 0.0000 |
>
> Even at high dimensionality, the total per-query overhead (softmax + projection + top-$k$) remains below 0.20 ms/query, demonstrating that the computational cost of PSP/LSP is negligible in practical retrieval pipelines.
>
> *Memory Footprint per Vector:*
>
> | Dim | Dense (KB/vec) | Sparse (KB/vec) | Reduction |
> |---:|:---:|:---:|:---:|
> | 1,024 | 4.00 | 1.00 | 4× (75.0%) |
> | 4,096 | 16.00 | 1.00 | 16× (93.8%) |
> | 16,384 | 64.00 | 1.00 | 64× (98.4%) |
> | 49,152 | 192.00 | 1.00 | 192× (99.5%) |
>
> The sparse representation stores only the top-$k$ ($k$ = 128) activated dimensions (index + value pair), yielding a fixed 1 KB/vector regardless of the original dimensionality. This is a direct consequence of the top-$k$ calibration property of the softmax function (Lapin et al., 2016; Yang \& Koyejo, 2020), which provides the theoretical justification for discarding low-ranked dimensions without losing discriminative information, as already validated empirically in Table 2 of the manuscript.
>
> As noted in Section 4.2.1, existing methods such as CoReS (Biondi et al., 2023a) require a pre-allocated fixed classifier whose memory footprint grows with the number of pre-allocated classes, plus an additional linear adaptation layer, making training infeasible at large scales. Regularization- and mapping-based methods (BCT, RACT, AdvBCT, LCE) operate on backbone encoder features whose dimensionality is fixed and independent of the number of classes, giving them a constant memory footprint per vector; however, they accumulate representation misalignment across successive updates (as shown in Tables 1 and 2) and require the old model during model updates. PSP and LSP, by contrast, require no regularized training, no mapping composition across updates, and achieve a low memory cost that is independent of the number of training classes.
>
> **[3. The paper does not discuss the case when the model has low accuracy]**
>
> We sincerely thank the Reviewer for this thoughtful and constructive suggestion, which helped us improve the paper by bringing out the connection between the result of Lemma A.1 (Eq. 13) and the accuracy of the model.
>
> We agree that when the model has low accuracy, softmax outputs and logits are less concentrated and will not cluster tightly around their class prototypes. Our theoretical analysis already makes this dependence explicit. Under the vMF model, the separation in expected cosine distance is scaled by the concentration terms. From Lemma A.1:
>
> $$\mathbb{E}\left[1 - X_1^\top X_2\right] = 1 - m_d(\kappa_1)\,m_d(\kappa_2)\,\mu_1^\top \mu_2,$$
>
> where $m_d(\kappa) \in (0,1)$ is monotonically increasing in $\kappa$ [Hornik & Grün (2014)]. When accuracy is low and representations are less concentrated (smaller $\kappa$), the factor $m_d(\kappa_1)m_d(\kappa_2)$ becomes small and expected distances move toward $\approx 1$ for both same-class and different-class pairs, reducing the separation required to satisfy Eq. 1b.
> Our result is therefore a conditional guarantee whose strength depends on the concentration of the learned representations.
>
>
> Importantly, Theorem 3.6 does not require high accuracy, but rather that the concentration parameter $\kappa$ of the updated model at step $t$ is greater that or equal to that of the old model at step $k$. This condition is empirically validated in Appendix H, which demonstrates under both the extended classes and the independently trained models scenarios that updated models yield more concentrated representations than their predecessors. The NC1 curves of Fig. 3 and Appendix C further confirm that concentration increases consistently throughout training toward convergence.
>
> `Action taken in the manuscript`: We added a specific paragraph in Section 3.4 addressing this observation.
>
> [Hornik & Grün (2014)] Hornik, Kurt, and Bettina Grün. "On maximum likelihood estimation of the concentration parameter of von Mises–Fisher distributions." Computational statistics 29.5 (2014): 945-957.

---

> ### Author Response · Authors · 2026-03-13
> **Response to Reviewer Mx5d (4/5)**
>
> **[4. Linear Probe setting to address the limitation of self-supervised model updates]**
>
> We thank the Reviewer for this constructive suggestion, which was also noted by Reviewer PRKR. We have conducted additional experiments to directly validate the integration of a linear probe on pretrained vision-language and self-supervised models.
>
> We evaluate PSP and LSP on two sequences of publicly available pretrained models (self-supervised and vision-language):
>
> - DINO-ViT-S → DINOv2-ViT-S/16 → DINOv3-ViT-L/16 (obtained from Hugging Face Hub)
> - CLIP ViT-B/32 → CLIP ViT-B/16 → CLIP ViT-L/14 (OpenAI pretrained models from the `open_clip` repository)
>
> For each model, a linear layer is trained on top of the extracted features using ImageNet-1K as the training set for 30 epochs, with encoder weights frozen. PSP and LSP features are derived from the softmax outputs and logits of this linear layer, respectively. We compare against the Baseline that uses the ℓ₂-normalized encoder output directly, without any additional training. Retrieval performance is then assessed on the ImageNet-1K test set.
>
> All metrics (AC, AA, ACA) follow the definitions in Section 4.1.
>
> ---
>
> Compatibility metrics for PSP and LSP features derived from a linear probe trained on ImageNet-1K. Bold indicates the highest value per metric group.
>
> | Method (DINO) | AC | AA | ACA |
> |:---|:---:|:---:|:---:|
> |  Baseline | 0.00 | 0.355 | 0.000 |
> |  LSP | 0.67 | 0.687 | 0.705 |
> |  PSP | **1.00** | **0.710** | **0.720** |
>
> | Method (CLIP) | AC | AA | ACA |
> |:---|:---:|:---:|:---:|
> |  Baseline | 0.00 | 0.307 | 0.000 |
> |  LSP | 0.33 | 0.521 | 0.493 |
> |  PSP | **1.00** | **0.601** | **0.592** |
>
> For DINO models, the observed trends are consistent with those reported for standard supervised models. PSP achieves AC = 1.0 (ACA = 0.72), while LSP achieves AC=0.67 (ACA = 0.705). This reflects the behaviour reported in Section 4.3 (Table 5), where LSP is more robust under distribution shift, and PSP excels in closed-set settings.
> The Baseline achieves reasonable same-model accuracy (diagonal values from 0.55 to 0.85), yet cross-model retrieval collapses to near zero in all conditions, with off-diagonal values as low as 0.0006. This suggests that independently pretrained self-supervised encoders produce representations that are not naturally aligned across updates.
>
> For CLIP models, we observe results following the same pattern (PSP: AC = 1.0, ACA = 0.59).
> The consistency of the PSP/LSP trade-off across both model families provides additional empirical support for the theoretical analysis in Sec. 3.4.
>
> `Action taken in the manuscript`: We have included the following compatibility matrices and the corresponding discussion in Appendix K of the revised manuscript. Accordingly, we revised the limitations to clarify that this issue can be mitigated through a linear-probe setting, which we now validate empirically.
>
> ---
>
> Below, we report the compatibility matrices for all Linear Probe experiments.
>
> Rows = Query model, Columns = Gallery model. Diagonal entries are same-model retrieval performance. Off-diagonal entries show cross-model retrieval performance, where ✓ = satisfies the compatibility criterion (Eq. 8); ✗ = does not.
>
> DINO (ViT-S/16 → DINOv2-S → DINOv3-ViT-L/16):
>
> - Baseline `AC=0.00 | AA=0.355 | ACA=0.000`
>
> | Query \ Gallery | DINO-ViT-S/16 | DINOv2-S | DINOv3-ViT-L/16 |
> |:---|:---:|:---:|:---:|
> | DINO-ViT-S/16 | 0.6371 | | |
> | DINOv2-S | 0.0006 ✗ | 0.6992 | |
> | DINOv3-ViT-L/16 | 0.0006 ✗ | 0.0006 ✗ | 0.7918 |
>
> - LSP `AC=0.67 | AA=0.687 | ACA=0.705`
>
> | Query \ Gallery | DINO-ViT-S/16 | DINOv2-S | DINOv3-ViT-L/16 |
> |:---|:---:|:---:|:---:|
> | DINO-ViT-S/16 | 0.6072 | | |
> | DINOv2-S | 0.5888 ✗ | 0.7033 | |
> | DINOv3-ViT-L/16 | 0.6769 ✓ | 0.7338 ✓ | 0.8104 |
>
> - PSP `AC=1.00 | AA=0.710 | ACA=0.720`
>
> | Query \ Gallery | DINO-ViT-S/16 | DINOv2-S | DINOv3-ViT-L/16 |
> |:---|:---:|:---:|:---:|
> | DINO-ViT-S/16 | 0.6191 | | |
> | DINOv2-S | 0.6590 ✓ | 0.6838 | |
> | DINOv3-ViT-L/16 | 0.7465 ✓ | 0.7536 ✓ | 0.7956 |
>
> ---
>
> CLIP (ViT-B/32 → ViT-B/16 → ViT-L/14):
>
> - Baseline `AC=0.00 | AA=0.307 | ACA=0.000`
>
> | Query \ Gallery | ViT-B/32 | ViT-B/16 | ViT-L/14 |
> |:---|:---:|:---:|:---:|
> | ViT-B/32 | 0.5508 | | |
> | ViT-B/16 | 0.0009 ✗ | 0.6020 | |
> | ViT-L/14 | 0.0009 ✗ | 0.0012 ✗ | 0.6856 |
>
> - LSP `AC=0.33 | AA=0.521 | ACA=0.493`
>
> | Query \ Gallery | ViT-B/32 | ViT-B/16 | ViT-L/14 |
> |:---|:---:|:---:|:---:|
> | ViT-B/32 | 0.4777 | | |
> | ViT-B/16 | 0.4498 ✗ | 0.5293 | |
> | ViT-L/14 | 0.4926 ✓ | 0.5244 ✗ | 0.6505 |
>
> - PSP `AC=1.00 | AA=0.601 | ACA=0.592`
>
> | Query \ Gallery | ViT-B/32 | ViT-B/16 | ViT-L/14 |
> |:---|:---:|:---:|:---:|
> | ViT-B/32 | 0.5361 | | |
> | ViT-B/16 | 0.5497 ✓ | 0.5931 | |
> | ViT-L/14 | 0.5932 ✓ | 0.6331 ✓ | 0.7002 |

---

> ### Author Response · Authors · 2026-03-13
> **Response to Reviewer Mx5d (5/5)**
>
> **[5. Models are also often updated to remove the impact of certain data points (unlearning) or to make them robust to new emerging threat models (robustness). These possibilities are not considered in the paper.]**
>
> We thank the Reviewer for broadening the scope of our manuscript to encompass a wider range of model update paradigms. However, the primary focus of our work is to demonstrate, both empirically and theoretically, that hyperspherical simplex representations derived from softmax outputs and logits are inherently backward-compatible. Given that backward-compatibility has a well-established literature, we deliberately restricted our analysis to consolidated update settings, with the exception of the backward-compatibility between independently trained models from public repositories presented in Sec. 4.3. We agree that extending backward-compatibility to unlearning and robustness to emerging threat models would be a valuable direction, and we consider it promising future work beyond the scope of this contribution.
>
> **[6. Broader Impact Concerns: There is no broader impact statement, and it would be nice to have one to discuss potential bias amplification resulting from proposed methods, where corrective updates for bias correction may not work as intended.]**
>
> Thank you for raising this point. We have not evaluated bias-related risks in this work, and we are not aware of prior work in the backward-compatibility literature that directly evaluates bias amplification effects under compatibility constraints.
>
> `Action taken in the manuscript`: We have added the following discussion to the paper after the Limitations:
> >While backward-compatible updates can reduce the cost of re-indexing, they may also preserve existing biases in legacy galleries, and the compatibility mechanism itself could introduce or amplify disparities if performance changes unevenly across subpopulations. Evaluating these effects and understanding when gallery refresh is required (e.g., for corrective debiasing updates) are important directions for future work.

---

### Review · Reviewer_PRKR · 2026-03-03

**Summary Of Contributions:**

The authors address the challenge of updating models while maintaining backward compatibility of representations as defined in Equations (1a) & (1b). The authors show theoretically that their two proposed methods (PSP and LSP) can satisfy the backward compatibility constraints in expectation. The key strength of the proposed method is that it does not require access to the previous model.

The key limitation is that it is currently not clear whether LSP satisfying backward compatibility is due to using different assumptions than Biondi et al. (2024) (see below for more details) or whether it is due to a property inherent to LSP.

Another limitation, acknowledged by the authors, is that the method seems to only apply to models trained using classification objectives, e.g., it does not directly apply to models like CLIP trained with contrastive losses. Although the authors briefly discuss a potential extension to other objectives at the end of the paper, it was not demonstrated.

**Audience:**

Yes

**Audience Explanation:**

Yes, the authors present a method for backward compatibility that works for independently trained models. Previous methods require fixing the classification head a priori. Thus if the company releasing the models do not adopt such an approach, practitioners that are fine-tuning the model for downstream tasks cannot easily integrate model updates.

**Claims And Evidence:**

No

**Claims Explanation:**

I have a question regarding the relationship between your theoretical results and those of Biondi et al. (2024).

Biondi et al. show that both conditions of backward compatibility (Eq. 1a and 1b) cannot be satisfied simultaneously, under the assumptions of Euclidean distance and uniform distributions over hyperballs. In contrast, your Theorem 3.6 shows that both conditions can be satisfied by LSP/PSP in expectation, under cosine distance and vMF distributional assumptions. Given this difference in assumptions, could you clarify to what extent the satisfiability of both equations is a consequence of the choice of distance metric and distributional model, rather than the specific use of normalized logits/softmax outputs (LSP/PSP)? For instance, would a similar result hold for normalized penultimate features under cosine distance and vMF assumptions?

Additionally, regarding the vMF assumption itself, the referenced work by Seo et al. (2021) demonstrates that penultimate layer activations can be statistically modeled by vMF distributions, rather than the logits (pre-softmax outputs) used in LSP. Since logits are obtained via a linear transformation of penultimate features followed by renormalization, it is not immediately obvious that the vMF property transfers to the logit space. Could you provide further justification or empirical validation for the vMF assumption specifically on the logit and softmax output spaces?

**Requested Changes:**

Page 4. [critical]:

Could the authors clarify the following statement: “In recent work, Biondi et al. (2024) empirically demonstrate that Eq. 1a is satisfied when clusters of features of the same class are aligned across model updates, while Eq. 1b is not satisfied.” In particular, why is Eq. 1b not satisfied? Providing intuition or a simple illustrative example would significantly improve clarity. Currently, it is unclear how the previous work was limited. A concise explanation would help readers understand the motivation of the proposed approach and technical contribution. See my question above, it is not clear whether LSP satisfies the constraints due to different assumptions on backward compatibility in Theorem 3.6 or whether it is due to properties intrinsic to LSP.

Page 5. [minor] small typo: “obtained applying the softmax function”

Page 5. [minor] Figure 1 is too confusing. Is a 3D figure necessary to convey the idea over a 2D figure? Moreover, it could be more suitable to split into multiple subfigures showing how you go from one space to another instead of overlaying all onto the same figure.

Page 6. [minor] The notation in Eq. 3 was not previously introduced $[v_c^k]_{c=1}^{C_k}$.

Page 7. [critical] I think $P_{t,k}$ is not square and therefore cannot be referred to as a valid projection matrix? (i.e. P^2 != P)

Page 8. [minor] Once again can these ideas be illustrated in 2D or is 3D necessary? Also is a fourth class in Figure. 4  needed? Maybe a simpler progression to comprehend would be:
two classes ⇒  three classes (unaligned)  ⇒ three classes (aligned)

Page 8. [critical]  The authors say “Prop. 3.4 establishes that the new class prototypes—and thus the angular misalignment $θ_c^{k→t
}$ —are orthogonal to the hyperplane spanned by the old prototypes.” How can “angular misalignment” be orthogonal to a prototype? Angular misalignment is scalar quantity and the prototype is a vector if I understand correctly. The wording seems to be  imprecise in this section.

Page 8. [minor] In Theorem 3.6, the separation between the assumptions (hypotheses) and the conclusion is not entirely clear. The statement would benefit from being rephrased with more explicit “if … then …” structure, so that the conditions under which the result holds and the resulting implication are more clear.

Page 13. [minor] In the extended backbone experiments i guess “t” is equal to 1? Should mention this and say the metrics are no longer really an average and are simply the accuracies.

---

> ### Author Response · Authors · 2026-03-13
> **Response to Reviewer PRKR (1/5)**
>
> We thank the Reviewer for their detailed and constructive evaluation of our manuscript. We are pleased that the Reviewer recognizes the key contributions of our paper, including the theoretical guarantees that PSP and LSP satisfy backward-compatibility constraints in expectation, and the practical significance of our approach in not requiring access to the previous model. We also appreciate the agreement that this is a practically important setting for the TMLR audience, particularly for practitioners who rely on externally released models and cannot easily integrate updates under existing approaches.
>
> In the following, we provide detailed responses to each of the requested changes and questions raised by the Reviewer.
>
> **[1. The relationship between our theoretical results and Biondi et al. (2024)]**
>
> The theoretical result of Biondi et al. (2024) and our Theorem 3.6 operate in different geometric settings. Biondi et al. (2024) analyze compatibility in Euclidean space using hyperball-distributed features. In their setting, squared Euclidean distance is unbounded, and inter-class separation depends on both cluster center displacement and cluster extent. Their Corollary 1 proves that both compatibility constraints cannot hold simultaneously under these specific assumptions.
>
> In contrast, our Theorem 3.6 operates in hyperspherical space with unit-norm features. Our representations lie on the unit hypersphere, where Euclidean distance is proportional to the cosine distance, and both metrics are therefore bounded, as cosine distance is defined in $[0, 2]$. Under this geometry, class separation is governed entirely by angular relationships between prototypes, rather than by unconstrained displacement and cluster extent as in the Euclidean setting of Biondi et al. (2024). This geometric difference is precisely the reason Eq. 1b cannot be satisfied in their setting but is satisfied analytically in expectation in ours. Their Corollary 1 establishes that both compatibility constraints cannot be simultaneously satisfied under squared Euclidean distance and hyperball-distributed features, making their result specific to that geometric setting.
>
>
> Biondi et al. (2024) state their theorem under a (d)-simplex fixed-classifier assumption, in which class centers are constrained to fixed, pre-allocated simplex positions. Their empirical evaluation of the expected distance uses a Monte Carlo procedure based on shifting a hyperball under a uniform assumption, which is a convenient numerical approximation but differs from the fixed-simplex configuration assumed in their theoretical statement. In contrast, our Theorem 3.6 keeps the simplex geometry explicit throughout the entire proof and yields a fully analytical derivation under the vMF model, without Monte Carlo integration.
>
> `Action taken in the manuscript:` We revised Section 3.1 to include the requested clarification from the Reviewer.
>
> **[2. Why normalized penultimate features fail under the same assumptions of distance metric and distributional model]**
>
> The satisfiability of both compatibility equations is a consequence of the specific use of normalized logits/softmax outputs (LSP/PSP) rather than the choice of distance metric and distributional model (further motivations can be found in the next point [3]).
>
> A similar result would not hold for normalized penultimate features, even under cosine distance and vMF assumptions. Satisfying Eq. 1a and Eq. 1b in expectation requires two strict geometric conditions: (i) class prototypes must be aligned across model updates, required for Eq. 1a, and (ii) the angle between prototypes of distinct classes must exceed $\pi/2$, required for Eq. 1b. Normalized penultimate features inherently fail the first condition. Even if such features were to converge to a Simplex ETF geometry during training (which can only occur when the embedding dimensionality is at least equal to the number of classes minus one), the Simplex ETF defines only the relative geometry between class prototypes, namely their angles and norms. It does not define their absolute orientation in the representation space. Two independently trained models can each converge to a Simplex ETF while having class prototypes pointing in entirely different absolute directions. Therefore, Simplex ETF convergence does not guarantee prototype alignment across model updates, and Eq. 1a would fail in expectation.

---

> > ### Comment · Reviewer_PRKR · 2026-03-18
> > **Further clarification on relationship to Biondi**
> >
> > Thank you for the clarification. I am still not certain your response answers this question:
> >
> > In the case of using a fixed d-simplex classifier as in Biondi et al (2024), would the use of normalized penultimate features lead to a similar result when using the same cosine distance and vMF distributional assumptions as in your Theorem 3.6?

---

> ### Author Response · Authors · 2026-03-13
> **Response to Reviewer PRKR (2/5)**
>
> **[3. How LSP/PSP intrinsically guarantees compatibility and its advantages over pre-allocated fixed classifiers]**
>
> The advantage of LSP/PSP is strictly tied to their derivation from the final model outputs. Softmax outputs and logits are explicitly optimized against one-hot encoded target labels. The canonical basis vectors $e_1, e_2, \dots$ constitute an absolute, fixed reference frame that does not vary across independent training runs. This provides three properties not achievable under the Simplex ETF assumption alone: (i) implicit alignment of same-class prototypes across model updates, (ii) exact quantification of the misalignment introduced by new classes via Eq. 6, and (iii) deterministic correction of that misalignment via $\mathbf{P}_{t,k}$, as proven in Prop. 3.5. The fixed reference frame allows our deterministic transformation to align representations across updates, satisfying Eq. 1a. The inherent regular simplex geometry additionally guarantees that the inter-class angle is always $\arccos(-1/(C-1)) > \pi/2$, satisfying Eq. 1b. These properties are intrinsic to the LSP/PSP construction and are not a consequence of the choice of distance metric or distributional model alone.
>
> We examine whether a method enforcing a pre-allocated d-simplex fixed classifier, such as CoReS (Biondi et al.,2023a), satisfies the same theoretical guarantees in practice. While such methods fix the simplex geometry, they have two fundamental limitations. First, unused pre-allocated prototypes generate negative gradients during loss optimization that interfere with the representation learning for actual classes (Fang et al., 2021), inducing misalignment between learned features and their designated fixed prototypes and undermining the alignment condition required by our proof. Second, the memory requirements grow substantially with the number of classes, driven by both the fixed classifier and the additional linear layer needed to adapt the model architecture to that classifier, making such approaches computationally prohibitive at scale, as directly evidenced by the infeasibility of training CoReS on Google Landmarks v2 reported in Tab. 2 of our paper. LSP/PSP avoids both issues entirely, achieving prototype alignment implicitly through the canonical basis construction, without any pre-allocated classifier, architectural modification, or prohibitive memory overhead.

---

> ### Author Response · Authors · 2026-03-13
> **Response to Reviewer PRKR (3/5)**
>
> **[4. vMF Assumption: Transfer to Logit and Softmax Spaces from Penultimate layer]**
>
> We thank the Reviewer for this observation. The role of the vMF($\mu, \kappa$) distribution as an assumption in our Theorem 3.6 is to provide a statistical modeling tool to analytically characterize the compatibility constraints. Specifically, the mean direction $\mu$ represents the class prototype of Eq. 2, while $\kappa$ controls the spread of features around that prototype. This modeling choice enables a closed-form analytical derivation of both compatibility inequalities, as demonstrated in Appendix A.
>
> Regarding the transfer of the vMF property from the penultimate layer to the logit and softmax spaces, we clarify that the vMF family is closed under orthogonal transformations such as rotations and reflections [Mardia & Jupp (2000); Romanazzi (2014)], but not under arbitrary linear maps. When a non-orthogonal transformation is applied, and the result is renormalized onto the sphere, the resulting directional distribution is in general anisotropic and better described by richer families such as the projected normal [Wang & Gelfand (2013)], the angular central Gaussian [Hernandez-Stumpfhauser et al. (2017)], or the Kent-Fisher-Bingham family [Kent (1982)]. To the best of our knowledge, there is no standard reference proving closure or non-closure for the exact transformation chain of our pipeline, namely a general linear map and normalization for logits, or a general linear map followed by the softmax function, centering and normalization for the softmax output space. We therefore do not claim that the vMF structure is automatically preserved under this transformation. Rather, the vMF assumption in the logit and softmax output spaces should be understood as a well-motivated modeling approximation, justified by the empirical evidence we provide below.
>
> We provide three complementary pieces of empirical validation directly in our paper:
>
> 1. Figure 3 and Appendix C directly measure the Neural Collapse hypotheses in both the logit and softmax output spaces across multiple architectures and datasets, confirming that representations in those spaces cluster tightly and symmetrically around their class prototypes at the terminal phase of training, which is consistent with a high-concentration vMF approximation $\kappa$.
> 2. Appendix H provides direct measurement of the mean cosine distance between features and their class prototypes across model updates, further confirming the increasing concentration consistent with the vMF assumption.
> 3. Our results in Section 4.3 demonstrate that PSP and LSP achieve backward-compatibility between independently trained models downloaded from public repositories, consistent with the vMF assumption holding in practice.
>
> We think that a formal theoretical analysis of the vMF transfer property is an interesting direction for future work.
>
> [Mardia & Jupp (2000)] Mardia, K. V., & Jupp, P. E. (2000). Directional Statistics. John Wiley & Sons.
>
> [Romanazzi (2014)] Romanazzi, M. (2014). Discriminant Analysis with High Dimensional von Mises-Fisher Distributions. Athens Journal of Sciences, 1(4), 225-240.
>
> [Wang & Gelfand (2013)] Wang, F., & Gelfand, A. E. (2013). Directional data analysis under the general projected normal distribution. Statistical Methodology, 10(1), 113-127.
>
> [Hernandez-Stumpfhauser et al. (2017)] Hernandez-Stumpfhauser, D., Breidt, F. J., & van der Woerd, M. J. (2017). The General Projected Normal Distribution of Arbitrary Dimension. Bayesian Analysis, 12(1), 113-133.
>
> [Kent (1982)] Kent, J. T. (1982). The Fisher-Bingham Distribution on the Sphere. Journal of the Royal Statistical Society: Series B, 44(1), 71-80.
>
> **[5. Small typo]**
>
> `Action taken in the manuscript:` We fixed the typo in the updated manuscripts.

---

> ### Author Response · Authors · 2026-03-13
> **Response to Reviewer PRKR (4/5)**
>
> **[6. Clarification on Figure 1]**
>
> We thank the Reviewer for this suggestion and clarify below the design choices behind Figure 1.
>
> The 3D representation is not a stylistic choice, but follows directly from the geometry of the hyperspherical simplex induced by the softmax outputs.
> Representations of $C$ classes naturally live in $\mathbb{R}^C$, where $C$ is the number of classes.
> In Figure 1, $C = 3$, so the probability simplex resides in $\mathbb{R}^3$, as also confirmed by Definition 3.2. Reducing to a 2D figure, for example, by adopting barycentric coordinates as in Figure 4, would eliminate the visible center misalignment between the original softmax outputs and the centered representations. This misalignment is precisely the geometric phenomenon that Figure 1 is designed to convey. The 3D representation is therefore essential to show that the probability simplex resides in the positive orthant of $\mathbb{R}^C$ and that centering shifts the simplex centroid to the origin, which is the key geometric step of our representations definition.
>
> We also wish to clarify a potential source of confusion regarding the Reviewer's suggestion to show the transition from one space to another. Our pipeline does not involve a change of space. The softmax outputs, the centered representations, and the final normalized representations all reside in the same space $\mathbb{R}^3$. Figure 1 illustrates successive geometric operations applied within that single space, namely centering followed by normalization onto the unit hypersphere. We hope this clarifies the Reviewer's concern.
>
> **[7.The notation in Eq. 3 was not previously introduced]**
>
> We thank the Reviewer for pointing this out and acknowledge that this notation was not explicitly introduced for Eq. 3. The bracket notation $[ \mathbf{v}^k_c \; ]_{c=1}^{C^k}$ denotes the matrix whose columns are the prototype vectors $\mathbf{v}^k_c$ for $c = 1, \dots, C^k$, i.e. the column-wise concatenation of all class prototype vectors of model at step $k$.
>
> `Action taken in the manuscript`: We added the following clarification in the revised manuscript directly after Eq. 3:
>
> >where $[ \mathbf{v}^k_c \; ]_{c=1}^{C^k}$ denotes the matrix obtained by stacking the class prototype vectors $\mathbf{v}^k_c$ as columns...
>
> **[8. $\mathbf{P}_{t,k}$ is not square and therefore cannot be referred to as a valid projection matrix]**
>
> We thank the Reviewer for this precise observation. The Reviewer is correct. $\mathbf{P}_{t,k} = \big[ \mathbf{V}^k | \mathbf{0} \big]$ is not square, meaning the idempotency condition $\mathbf{P}^2 = \mathbf{P}$ cannot hold. We apologize for this terminological inaccuracy.
>
> To be precise, $P_{t,k}$ is a deterministic linear transformation, independent of any learnable parameters or model forward passes, composed of two parts: (a) the square projection matrix $\mathbf{V}^k = \mathbf{P}_{k,k}$, which is a valid orthogonal projection satisfying $(\mathbf{V}^k)^2 = \mathbf{V}^k$, acting on the first $C^k$ dimensions, and (b) the zero matrix $\mathbf{0}$ of dimension $C^k \times (C^t - C^k)$, which discards the dimensions introduced by the newly added classes.
>
> `Action taken in the manuscript`: We corrected the terminology in the revised manuscript accordingly, replacing references to $\mathbf{P}_{t,k}$ as a projection matrix with the more precise term: deterministic linear transformation.
>
>
> **[9. Simplification of Figure 4]**
>
> We thank the Reviewer for this suggestion. We wish to clarify that Figure 4 already addresses the dimensionality concern raised for Figure 1. Specifically, Figure 4 adopts barycentric coordinates, which reduce the representation from $\mathbb{R}^C$ to a $(C-1)$-dimensional simplex. This allows the focus to remain on the evolution of class prototypes and their misalignment across model updates.
>
> Regarding the Reviewer's suggestion to simplify the progression to two and then three classes, we respectfully note that $C = 4$ is the last case that can be visually represented, since barycentric coordinates reduce the 4-class simplex to a 3D tetrahedron, which is the highest-dimensional case that remains human-interpretable. Limiting the figure to two and then three classes would restrict the illustration to special low-class regimes and would not convey that the misalignment and alignment phenomena are general properties that hold at every incremental update step, regardless of the number of classes. The progression shown in Figure 4, from $C^k$ to $C^t = C^k + 1$ classes in the unaligned case and then to the aligned case, is designed to illustrate precisely this generality. We believe the current figure effectively conveys this, but we are happy to simplify it if the Reviewer thinks that is strictly necessary.

---

> ### Author Response · Authors · 2026-03-13
> **Response to Reviewer PRKR (5/5)**
>
> **[10. How can “angular misalignment” be orthogonal to a prototype]**
>
> We thank the Reviewer for this precise observation and apologize for the imprecise wording. The Reviewer is correct that the angular misalignment $\theta_c^{k\to t}$ is a scalar quantity and therefore cannot be orthogonal to a prototype.
>
> The precise geometric statement is the following:
> >Prop. 3.4 establishes that the prototype vectors of the newly added classes at step $t$ are orthogonal to the hyperplane spanned by the old prototypes at step $k$. As a direct consequence of this orthogonality, when the simplex expands to accommodate the new classes, the existing class prototypes acquire nonzero components along these new orthogonal dimensions at step $t$, inducing the angular displacement quantified as $\theta_c^{k\to t}$ in Eq. 6.
>
> `Action taken in the manuscript`: We corrected the imprecise sentence in the revised manuscript accordingly.
>
> **[11. Separation between the assumptions (hypotheses) and the conclusion in Theorem 3.6]**
>
> We thank the Reviewer for this suggestion. We agree that making the hypothesis-conclusion structure more explicit improves clarity.
>
> `Action taken in the manuscript`: We rephrased Theorem 3.6 in the revised manuscript to have a better logical separation between the assumptions and the conclusion.
>
> **[12. Clarification on the extended backbone experiments]**
>
> The Reviewer is correct that in Section 4.2.2 (Extended Backbone Benchmark) and Section 4.2.3 (Extended Classes and Backbone Benchmark), both reported in Table 4, there is only a single update step. The metrics AC, AA, and ACA are defined as averages over the compatibility matrix entries and remain valid in this setting. AC and AA are summary metrics introduced by Biondi et al. (2023a) to aggregate the entries of the compatibility matrix $\mathbf{C}_{t,k}$ across update steps.
>
> In these experiments, the first model corresponds to step $t = 1$ and the new model to step $t = 2$, giving $T = 2$. In this setting, AC reduces to a single compatibility check $(C_{2,1} >C_{1,1})$, AA is the average of three matrix entries, and ACA is the cross-model performance value when compatibility is satisfied, and to 0 otherwise.
>
> `Action taken in the manuscript`: An explicit clarification of the number of steps in these two experiments has been added to the revised manuscript.
>
> **[13. Addressing the limitation of applicability to contrastive and self-supervised models]**
>
> In the limitations paragraph of our conclusion section, we acknowledged that PSP and LSP require models to be trained using classification as a surrogate task, since the alignment of features to a regular simplex structure relies on the optimization dynamics of the softmax cross-entropy loss with one-hot encoded target labels. We also noted that this limitation could be mitigated by training a linear classification head on top of a frozen pretrained encoder. As also requested by Reviewer Mx5d, we have now directly validated it by conducting additional experiments on sequences of publicly available self-supervised (DINO family: ViT-S to ViT-L) and vision-language (CLIP family: ViT-B/32 to ViT-L/14) pretrained models. A linear layer (without the bias term) is trained on top of the frozen encoder, and PSP and LSP features are derived from its softmax outputs and logits. The results, included in Appendix K of the revised manuscript, demonstrate that PSP and LSP achieve consistent backward-compatibility on DINO and CLIP models. We refer the Reviewer to our detailed response to Reviewer Mx5d, point [4], for the full experimental setup, compatibility matrices, and discussion of results.
>
> `Action taken in the manuscript`: We have included the compatibility matrices and the corresponding discussion in Appendix K of the revised manuscript. Accordingly, we revised the limitations to clarify that this issue can be mitigated through a linear-probe setting, which we now validate empirically.

---

> > ### Comment · Reviewer_PRKR · 2026-03-22
> > **Angular misalignment**
> >
> > Thank you for the revised wording. However, the statement "the existing class prototypes acquire nonzero components along these new orthogonal dimensions at step t, inducing the angular displacement" reads as though the angular displacement is solely due to the new dimensions having nonzero values. From my understanding, adding new classes shifts the centroid, which redefines the prototypes in all components, not just the newly added dimensions?

---

> > > ### Comment · Reviewer_PRKR · 2026-03-22
> > > **Angles between vectors of different dimension**
> > >
> > > The wording in Section 3.3 (specifically Proposition 3.4 and the paragraph following it) is not very precise. Propositions 3.4 and 3.5 involve comparing prototypes across different update steps, where the prototypes live in different dimensional spaces ($\mathbf{v}_c^k \in \mathbb{R}^{C^k}$ and $\mathbf{v}_c^t \in \mathbb{R}^{C^t}$). Thus statements about orthogonality and angular misalignment between these vectors are confusing. I believe the intention is to embedding the lower-dimensional prototypes into $\mathbb{R}^{C^t}$ via zero-padding and thereafter comparing against $\mathbf{v}_c^t$. But this is currently left implicit for the reader to infer.

---

> > > > ### Author Response · Authors · 2026-03-23
> > > > **Replying to "Angles between vectors of different dimension"**
> > > >
> > > > We thank the Reviewer for this observation. We agree that, in the previous version, the comparison between prototypes from different update steps was not sufficiently explicit in the statements of the propositions and in the surrounding main text. In particular, the zero-padding step used to embed prototypes from old steps into the higher-dimensional space was left implicit for the reader in Section 3.3, even though it was already explicitly stated in the proofs. Our original intention was not to make this explicit in the main text, as we regarded it as a purely auxiliary operation, introduced solely in the proof to compute inner products across steps.
> > > >
> > > > `Action taken in the revised manuscript`:
> > > > To remove this ambiguity while avoiding excessive notational clutter in the mathematical statements, we have chosen not to alter the text of Propositions 3.4 and 3.5 themselves. Instead, we have added a clarifying paragraph immediately preceding the propositions.
> > > > The revised text reads:
> > > > > The following two propositions formalize the properties of $\mathbf{P}_{t,k}$, where comparisons across update steps are performed after embedding the lower-dimensional prototypes into the higher-dimensional space via zero-padding.

---

> ### Author Response · Authors · 2026-03-18
>
> We thank the Reviewer for this follow-up. The answer is yes: under the same assumptions used in our analysis, a $d$-simplex fixed classifier as in Biondi et al. (2024) would satisfy the compatibility inequalities of Eq. 1a and Eq. 1b in expectation, provided that its regular simplex geometry remains aligned across model updates.
>
> This broader setting is closely related to our analysis. In the paper, however, we chose to state the theorem for the specific representations introduced in this work, namely PSP and LSP, which are constructed from softmax outputs and logits, rather than for the more general class of aligned regular simplex-based representations.
>
> This distinction matters from a modeling perspective. Fixed simplex classifiers require the classifier geometry to be imposed during training through a pre-allocated set of prototypes, which introduces a dependency across models, since the same fixed design must be shared in advance. In contrast, our setting studies compatibility properties of representations derived from softmax outputs and logits, and therefore does not require a classifier geometry to be specified across parties. As a result, our method can achieve compatibility even between independently trained models from public repositories, as shown in Section 4.3 of the manuscript. For these reasons, we focused the theorem on PSP and LSP, though we now make the extension explicit in the Remark after the compatibility proof.
>
> `Action taken in the revised manuscript:`
> To make this connection explicit, we added the following remark in the appendix after the compatibility proof:
>
> > **Remark** (*Extension to aligned regular simplex representations*). Under the same assumptions of Theorem 3.6, the compatibility result extends to any regular simplex representation whose geometry remains aligned across model updates. In particular, this includes $d$-simplex fixed classifiers such as those considered by Biondi et al. (2023a; 2024).

---

> > ### Comment · Reviewer_PRKR · 2026-03-22
> > **Further clarification on relationship to Biondi (2)**
> >
> > Thank you for the clarification and for making the update in the appendix. However, given that under that Biondi et al's fixed classifier would also satisfy the compatibility equations under the distributional assumption (vMF), I would say this needs to be made more clear in the main text. The main contribution of the PSP/LSP methods is that it enables the compatibility equations to be satisfied without having to fix the classifier heads a priori (under vMF). Biondi et al also satisfies the compatibility equations (under vMF).
> >
> > However, as is currently written Section 3.1 is misleading, and implies that your method satisfies the compatibility equations whereas Biondi at al does not. Moreover, the details added by the authors in this section during the discussion perior regarding the Biondi et al method do not clarify this  issue. In particular, they will be difficult for a reader to comprehend without having full grasped the theoretical results of both papers.
> >
> > Perhaps re-writing this section with a focus on your approach not requiring a fixed a classifier head compared with Biondi would be more faithful. Moreover, the main surrounding the theorem presented should note the difference in assumptions in the theorem (maybe with a simple reference to a section in the appendix to clarify this)

---

> > > ### Author Response · Authors · 2026-03-24
> > > **Replying to "Further clarification on relationship to Biondi (2)"**
> > >
> > > We thank the Reviewer for this important suggestion. We agree that this distinction needed to be made more explicit in the main text. To address this point, we revised the manuscript in both Sec 3.1 and Sec 3.4.
> > >
> > > First, in Sec. 3.1, we rewrote the comparison with Biondi et al. to make the distinction clearer. In particular, we now clarify that the conclusion of Biondi et al is derived under a Euclidean hyperball formulation, where their analysis establishes the same-class compatibility condition (Eq. 1a) but not the different-class one (Eq. 1b). We then state explicitly that, under the hyperspherical vMF assumption used in our theorem, the compatibility result is not exclusive to PSP/LSP, but also extends to regular simplex representations that remain aligned across model update, such as the one learned through $d$-simplex fixed classifiers considered by Biondi et al. In other words, the revised text now makes clear that the key distinction is not that Biondi et al.’s $d$-simplex fixed classifier representations would fail under these assumptions, but rather that PSP/LSP satisfy the compatibility conditions without requiring the classifier head to be fixed a priori.
> > >
> > > Second, in the text surrounding Theorem 3.6 in Sec. 3.4, we added a dedicated Discussion paragraph to make the assumptions and scope of the theorem explicit in the main paper. There, we clarify that the theorem is derived under a hyperspherical von Mises-Fisher model with cosine-distance analysis and increasing concentration across updates, and we explain how these assumptions differ from those in Biondi et al., whose analysis is done in a Euclidean hyperball formulation. In the same discussion, we also added an explicit pointer to Remark 1 in the appendix to indicate that, under our assumptions, the theorem extends more generally to any regular simplex representations that remain aligned across model updates.

---

> ### Author Response · Authors · 2026-03-23
> **Replying to "Angular misalignment"**
>
> We thank the Reviewer for this clarification and agree that the angular misalignment is not caused solely by the nonzero components in the newly introduced orthogonal dimensions. When new classes are added, the expansion of the probability simplex shifts its center, which in turn redefines the existing class prototypes in all dimensions.
>
> `Action taken in the revised manuscript:`
> We corrected the imprecise sentence in the revised manuscript accordingly.
>
> > Prop. 3.4 establishes that the prototypes of the newly added classes at step $t$ are orthogonal to the hyperplane spanned by the old prototypes at step $k$. When the probability simplex is expanded to accommodate new classes, its center shifts, thereby redefining the existing class prototypes in all dimensions. As a result, the updated prototypes are no longer confined to the old hyperplane and acquire nonzero components along its orthogonal directions. This induces the angular misalignment $\theta_c^{k\to t}$ defined in Eq. 6.

---

### Author Response · Authors · 2026-03-13
**Summary of Changes in the Revised Manuscript**

We sincerely thank the Action Editor and all Reviewers for their careful reading of our manuscript, and for their constructive comments and suggestions.

We have uploaded a revised version of the manuscript, where all changes introduced after revision are highlighted in **blue**.

The main revisions are summarized below:
- clarification of the proposed method and its practical integration in retrieval settings;
- improved presentation of the theoretical assumptions, statements, and guarantees;
- additional experiments further validating the applicability and scalability of the proposed approach on self-supervised pretrained model.

We hope that these revisions have addressed the concerns raised and improved the clarity and overall quality of the manuscript.

---

### Author Response · Authors · 2026-03-17
**Follow-up on the Discussion Period**

Thank you again for your time and thoughtful feedback on our manuscript.

As the discussion period deadline is approaching, we wanted to kindly follow up and say that we are looking forward to further discussion with the reviewers, as this exchange is very valuable for improving the paper.

Thank you again for your effort and consideration.

---

### Decision · Action_Editor_y6wH · 2026-05-29

**Recommendation:** Reject

**Audience:**

Yes

**Audience Explanation:**

The paper studies the problem of updating models representation continually, specifically, how to insure backward compatibility for information retrieval of previously indexed vectors without reindexing.

**Claims And Evidence:**

No

**Claims Explanation:**

Reviewer PRKR has raised an important concern about inaccurate claim in the paper "that the fixed d-simplex classifier of Biondi et al (2024) could not satisfy the compatibility equations whereas their approach does". The author-reviewer discussions did not help completely clear the reviewer concern and the modifications made to the paper seem to fail to accurately articulate the uniqueness of the approach and conditions on which the claim can be made. With this, I suggest the authors to rewrite their claims and revise the paper and submit again.

**Resubmission Of Major Revision:**

The authors may consider submitting a major revision at a later time.

---

> ### Author Response · Authors · 2026-06-04
> **Comment to the Final Decision**
>
> Dear Action Editor,
>
> Thank you for the decision. We understand the concern about inaccurate claim in the paper. In particular, we now see that the wording in Section 3.1 could be read as making a method-level contrast between Biondi et al. and our approach, especially around the sequence “Eq. 1b is not” followed by “In contrast, we…”. This was not our intended claim.
>
> The intended distinction was between analytical settings, not a claim that fixed-simplex methods fail under the assumptions of our theorem. Under the same aligned regular-simplex/vMF assumptions, with increasing concentration across updates, fixed d-simplex representations such as those considered by Biondi et al. also fall within the scope of our result and can satisfy the compatibility equations in expectation. In this sense, our analysis provides a complementary closed-form hyperspherical explanation for why such simplex-stationary representations can be compatible.
>
> The contribution we intended to emphasize is that the closed-form expected cosine-distance analysis makes explicit the angular condition under which Eq. 1b holds, and that PSP/LSP obtain such simplex-compatible representations from logits or softmax outputs without requiring classifier heads to be fixed or pre-allocated a priori.
>
> We recognize from the decision and reviewer discussion that this distinction was not sufficiently integrated into the main framing of the paper. We will revise the claims accordingly, making the scope of the theorem and the relation to fixed-simplex classifiers explicit from the beginning.
>
> Best regards,
> The Authors